# Plasma Tie2 is a tumor vascular response biomarker for VEGF inhibitors in metastatic colorectal cancer

Gordon C. Jayson[1], Cong Zhou[2], Alison Backen[3], Laura Horsley[1], Kalena Marti-Marti[1], Danielle Shaw[4], Nerissa Mescallado[1], Andrew Clamp[5], Mark P. Saunders[5], Juan W. Valle[1], Saifee Mullamitha[5], Mike Braun[5], Jurjees Hasan[5], Delyth McEntee[5], Kathryn Simpson[3], Ross A. Little[6], Yvonne Watson[6], Susan Cheung[6], Caleb Roberts[6], Linda Ashcroft[5], Prakash Manoharan[1], Stefan J. Scherer[7], Olivia del Puerto[8], Alan Jackson[6], James P.B. O'Connor[2], Geoff J.M. Parker[6,9] & Caroline Dive[3]

Oncological use of anti-angiogenic VEGF inhibitors has been limited by the lack of informative biomarkers. Previously we reported circulating Tie2 as a vascular response biomarker for bevacizumab-treated ovarian cancer patients. Using advanced MRI and circulating bio-markers we have extended these findings in metastatic colorectal cancer ($n = 70$). Bevacizumab (10 mg/kg) was administered to elicit a biomarker response, followed by FOLFOX6-bevacizumab until disease progression. Bevacizumab induced a correlation between Tie2 and the tumor vascular imaging biomarker, $K^{trans}$ ($R$: −0.21 to 0.47) implying that Tie2 originated from the tumor vasculature. Tie2 trajectories were independently associated with pre-treatment tumor vascular characteristics, tumor response, progression free survival (HR for progression = 3.01, $p = 0.00014$; median PFS 248 vs. 348 days $p = 0.0008$) and the modeling of progressive disease ($p < 0.0001$), suggesting that Tie2 should be monitored clinically to optimize VEGF inhibitor use. A vascular response is defined as a 30% reduction in Tie2; vascular progression as a 40% increase in Tie2 above the nadir. Tie2 is the first, validated, tumor vascular response biomarker for VEGFi.

[1] The Christie NHS Foundation Trust and Division of Cancer Sciences, University of Manchester, Manchester M20 4BX, UK. [2] Division of Cancer Sciences, Manchester Cancer Research Centre, University of Manchester, Manchester M20 4GJ, UK. [3] Clinical and Experimental Pharmacology Group, Cancer Research UK Manchester Institute & Manchester Centre for Cancer Biomarker Sciences, Manchester M20 4BX, UK. [4] Clatterbridge Cancer Centre, Liverpool CH63 4JY, UK. [5] Manchester Academic Health Science Centre, Trials Co-ordination Unit, The Christie NHS Foundation Trust, Withington Hall Block C, Wilmslow Road, Manchester M20 4BX, UK. [6] Imaging Sciences, University of Manchester, Manchester M13 9PT, UK. [7] Novartis Pharmaceuticals Corporation, One Health Plaza, 337, East Hanover, NJ 07936-1080, USA. [8] Del Puerto Limited, 23 Porters Wood; Saint Albans, Hertfordshire AL3 6PQ, UK. [9] Bioxydyn Ltd, Manchester M15 6SZ, UK. These authors contributed equally: Gordon C. Jayson, Cong Zhou. These authors jointly supervised to this work: James P. B. O'Connor, Geoff J. M. Parker, Caroline Dive. Correspondence and requests for materials should be addressed to G.C.J. (email: Gordon.Jayson@manchester.ac.uk)

Anti-angiogenic VEGF inhibitors (VEGFi) have become some of the most widely prescribed drugs in oncology[1]. Despite intense effort to find biomarkers that would optimize treatment with VEGFi, predictive biomarkers have proven elusive. However, there have been few attempts identify and validate response biomarkers. Several trials[2,3] have reported pharmacodynamic changes in a variety of circulating biomarkers during treatment with VEGFi. The most consistent pharmacodynamic data pertain to the plasma concentration of the angiopoietin receptor, Tie2. The findings are of considerable interest as (i) they are mechanistically plausible given the interaction between VEGF and Angiopoietin signaling pathways;[4,5] (ii) VEGFi-induced pharmacodynamic changes in Tie2 have been observed in glioma[2], gall bladder[6], and colorectal cancer;[3] and (iii) the Ang/Tie2 pathway is a known and valid clinical target in colorectal[7] and ovarian cancer[8–10]. Despite these important associations, the clinical significance of Tie2 behavior has not been determined.

Recently, we completed studies in ovarian cancer, which revealed that Tie2 functions as a vascular response biomarker in patients treated with the anti-VEGF antibody, bevacizumab, and cytotoxic therapy[11,12]. In these studies, we showed that bevacizumab-induced PD changes in Tie2 were of clinical value. In particular, progression modeling was significantly improved when Tie2 (the vascular response biomarker) was studied with CA-125 (the epithelial marker). While these ovarian cancer studies provided insight into the potential clinical utility of Tie2 as a vascular biomarker they were based solely on circulating biomarkers and lacked direct tumor-specific data that can only be generated through imaging. Here, we deployed advanced imaging alongside the circulating biomarkers to qualify, in metastatic colorectal cancer (mCRC), Tie2 as the first, generic, tumor vascular response biomarker for VEGFi. In addition the study validates the concept that a cancer consists of multiple tissue types, in this case the epithelial and vascular components, which should be considered as targets at each recurrence of cancer treatment; the multi-tissue compartment model of cancer therapy.

## Results

Seventy patients with mCRC were recruited. Their demographic and survival statistics are shown in Table 1 and Fig. 1. In all, nineteen circulating biomarkers and nine imaging biomarkers were measured (Table 2 and Supplementary Figure 1). The technical variation of each circulating biomarker was estimated using technical replicates (Supplementary Figure 2). Most biomarkers demonstrated technical variation within the range 76–132% (95% CI). Pre-treatment intra-patient variations of biomarkers demonstrated a range of repeatability values (Supplementary Figure 3).

**Pre-treatment biomarker networks are prognostic**. The study investigated a significant number of biomarkers. Therefore, we performed an initial analysis to evaluate the relationship between all the biomarkers with one another to avoid the confounding factor of multiple comparisons while simultaneously focusing attention on key sets of biomarkers. To do this, we examined the change in correlative relationships between biomarkers before and during treatment. Pearson correlation networks, as described previously[11,12], of the circulating and imaging biomarkers were generated to determine the presence of positive or negative correlations (Fig. 2).

Before treatment many circulating biomarkers that relate to the biology of angiogenesis formed strongly correlated clusters (median $r = 0.40$, $r$ ranges from $-0.21$ to $0.97$). This infers

**Table 1 Baseline patient characteristics**

|  | N | Percentage or range |
|---|---|---|
| Total patients | 70 | 100 |
| Male | 41 | 59 |
| Female | 29 | 41 |
| Age (median) | 63 years | 29–77 |
| ECOG Performance Status 0 | 38 | 54 |
| ECOG Performance Status 1 | 29 | 41 |
| ECOG Performance Status 2 | 3 | 4 |
| Disease site—bowel | 58 | 83 |
| Disease site—liver | 58 | 83 |
| Disease site—lung | 25 | 36 |
| Disease site—lymph nodes | 24 | 34 |
| Disease site—bone | 1 | 1 |
| Disease site—other | 10 | 14 |
| 1 metastatic site | 31 | 45 |
| 2 metastatic sites | 26 | 37 |
| 3 + metastatic sites | 13 | 18 |
| Well differentiated tumor | 7 | 10 |
| Moderately differentiated tumor | 45 | 64 |
| Poorly differentiated tumor | 9 | 13 |
| Unknown degree of differentiation | 9 | 13 |
| Primary tumor in situ—yes | 22 | 31 |
| Primary tumor in situ—no | 48 | 69 |
| Platelets (normal range 150–400) | 373 | $172–931 \times 10^9$/L |
| LDH (normal range < 450) | 475 | 157–9335 IU/L |
| CEA (normal range < 5) | 47 | <3–16,485 IU/L |

*CEA* carcino-embryonic antigen, *ECOG* Eastern Cooperative Group, *IU* international units, *LDH* lactate dehydrogenase, *L* litre

biologically coordinated regulation of the neo-vasculature (Fig. 2a)[12]. Imaging biomarkers of vascular function, IAUC$_{60}$, $K^{trans}$, $v_p$ and Enhancing Fraction (EF) also correlated strongly with each other (Fig. 2a). However, there was no evidence of correlations between circulating and imaging biomarkers. This implies that in treatment-naive patients, the circulating and imaging biomarkers measured different aspects of tumor neo-vasculature.

Further, the biomarker of tumor burden, circulating CK18[13,14], which relates to epithelial cell death, and Whole Tumor Volume (WTV) correlated strongly with one another. However, these biomarkers did not correlate with biomarkers of the neo-vasculature (Fig. 2a). This indicates a biological and functional distinction between biomarker signatures that represent the epithelial and vascular tissue compartments.

Next, we explored the associations between pre-treatment clinical and demographic factors and the circulating and imaging biomarkers. However, the only significant association to emerge from this part of the analysis was between total tumor volume and performance status (PS, $p = 0.0078$, chi-squared tests). Univariate analysis of pre-treatment clinical, circulating and imaging biomarkers for their association with progression free survival (PFS) indicated that PS, LDH, VEGF-A, VEGF-R2, CK18, total tumor volume, IAUC, $K^{trans}$, WTV, ETV, and EF were potential prognostic biomarkers. To investigate further, we applied a multivariate Cox proportional hazard regression analysis supported by a bootstrap resampling approach for validation. This method examines the stability of the findings when different data points are included or excluded from the analysis. Through this, we identified performance status, total tumor volume and the ratio of the plasma concentration of VEGF-R2 to tumor $K^{trans}$ as pre-treatment independent prognostic biomarkers for PFS (Fig. 2b, Table 3). The key finding was that patients with higher VEGF-R2 to tumor $K^{trans}$ ratios had a worse PFS.

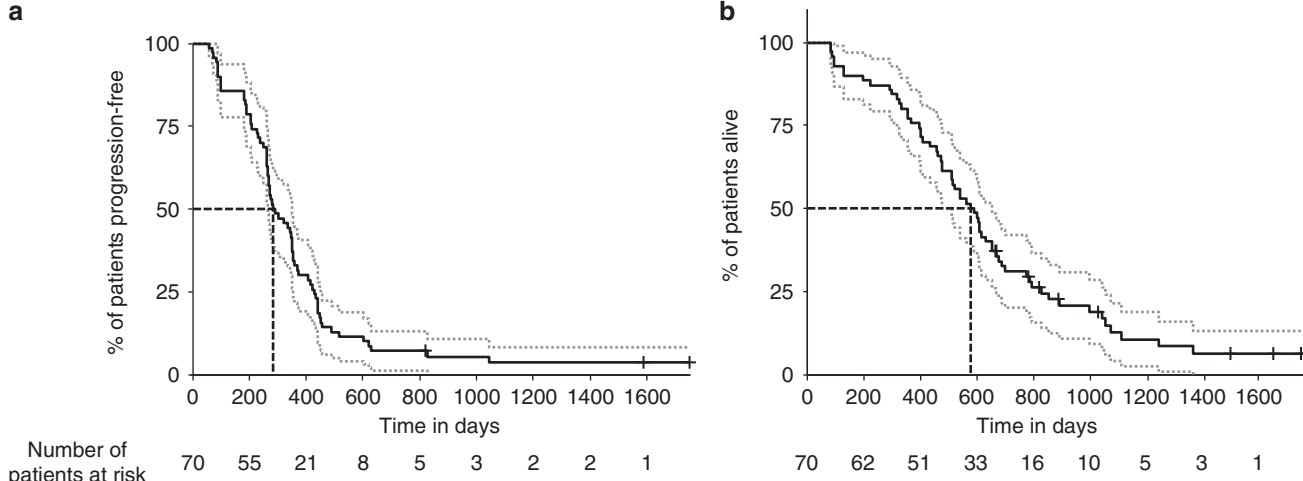

**Fig. 1** Progression-free and overall survival Kaplan–Meier estimator graphs. **a** Shows the proportion of the 70 eligible patients who are alive and free from disease-progression; the median PFS was 283 days (9 months; 95% CI 265–351 days or 8.8–11.7 months). One year and 2 year PFS statistics were 31 and 7% respectively. Note; no patient developed PD between days 98 and 182. **b** Shows the proportion of the 70 eligible patients who are alive; the median OS was 578 days (19.3 months; 95% CI 477–651 days or 15.9–21.4 months). One and 2-year survival statistics were 76 and 29%, respectively. The slightly lower OS length are not statistically shorter than other published series, but likely reflect the eligibility requirement for at least one metastatic lesion with a diameter between 3 and 10 cm that rendered the lesion amenable to serial MRI. Solid black lines; progression-free or overall survival, dotted gray lines; 95% CI, dashed black lines; median values, black crossed; patients whose data was censored. PFS progression-free survival, OS overall survival, CI confidence interval, MRI magnetic resonance imaging

| **Table 2 Summary of baseline biomarker levels and changes during treatment** | | | | | |
|---|---|---|---|---|---|
| **Biomarker type** | **Biomarker name** | **Pre-treatment levels (Median ± SD)** | **Median change at 14 days with single agent bevacizumab (%)** | **Biggest median change during treatment (%)** | **Median change at disease progression (%)** |
| Circulating | Ang1 | $2.5 \pm 2.0 \times 10^3$ | 102.9 | 50.8 | 50.8 |
| Circulating | Ang2 | $4.9 \pm 9.0 \times 10^2$ | 76.1 | 65.7 | 72.5 |
| Circulating | E-selectin | $1.5 \pm 0.9 \times 10^4$ | 92.8 | 86.9 | 96.4 |
| Circulating | FGF2 | $2.6 \pm 13.0 \times 10^2$ | 86.5 | 46.9 | 57.5 |
| Circulating | HGF | $4.2 \pm 12.2 \times 10^2$ | 93.1 | 82.0 | 95.7 |
| Circulating | IL6 | $1.1 \pm 6.7 \times 10^1$ | 104.3 | 148.3 | 148.3 |
| Circulating | IL8 | $5.0 \pm 9.6 \times 10$ | 88.6 | 71.7 | 87.1 |
| Circulating | KGF | $7.3 \pm 68.4 \times 10^0$ | 92.3 | 41.1 | 106.7 |
| Circulating | CK18 | $1.1 \pm 0.6 \times 10^3$ | 91.5 | 58.1 | 86.7 |
| Circulating | PDGFbb | $2.4 \pm 2.2 \times 10^2$ | 97.2 | 70.6 | 82.5 |
| Circulating | PLGF | $3.3 \pm 41.0 \times 10^1$ | 104.1 | 92.9 | 113.3 |
| Circulating | SDF1b | $1.8 \pm 1.2 \times 10^3$ | 103.6 | 114.1 | 101.3 |
| Circulating | Tie2 | $1.4 \pm 0.8 \times 10^4$ | 84.4 | 80.8 | 84.8 |
| Circulating | VCAM1 | $5.6 \pm 1.9 \times 10^5$ | 103.6 | 186.3 | 183.6 |
| Circulating | VEGFA | $1.9 \pm 1.9 \times 10^2$ | 74.8 | 56.7 | 62.0 |
| Circulating | VEGFC | $5.8 \pm 10.0 \times 10^2$ | 94.8 | 64.6 | 81.4 |
| Circulating | VEGFD | $5.6 \pm 255.3 \times 10^2$ | 100.9 | 87.9 | 94.2 |
| Circulating | VEGFR1 | $6.2 \pm 78.7 \times 10^1$ | 96.0 | 91.6 | 115.7 |
| Circulating | VEGFR2 | $1.1 \pm 0.7 \times 10^4$ | 97.6 | 84.0 | 84.0 |
| Imaging | ADC | $1.0 \pm 0.2 \times 10^{-3}$ | 102.4 | 91.0 | 98.8 |
| Imaging | EF | $9.7 \pm 0.5 \times 10^1$ | 96.8 | 95.9 | 97.1 |
| Imaging | ETV | $3.8 \pm 12.8 \times 10^4$ | 84.4 | 43.6 | 48.3 |
| Imaging | WTV | $3.9 \pm 13.9 \times 10^4$ | 89.6 | 39.9 | 61.1 |
| Imaging | IAUC | $1.7 \pm 0.7 \times 10^1$ | 72.1 | 68.0 | 72.9 |
| Imaging | $K^{trans}$ | $1.7 \pm 1.1 \times 10^{-1}$ | 71.7 | 66.5 | 74.1 |
| Imaging | T1 | $1.1 \pm 0.2 \times 10^3$ | 94.0 | 86.8 | 90.4 |
| Imaging | Ve | $3.1 \pm 1.1 \times 10^{-1}$ | 88.7 | 86.5 | 86.5 |
| Imaging | Vp | $1.7 \pm 1.6 \times 10^{-2}$ | 67.8 | 63.4 | 63.4 |

Median pre-treatment levels of circulating and imaging biomarkers are shown, with the standard deviation (SD). Changes through treatment at three time-points are shown, related as a relative percentage (%) to the pre-treatment baseline levels. Values above 100 indicate increased biomarker levels while values below 100 indicate reduced biomarker levels. Plots of biomarker changes are shown in Supplementary Figure 2

The units for circulating biomarkers are pg/ml. The units of ADC are mm2/s; ETV and WTV mm3; IAUC s.mmol/l; Ktrans 1/min; T1 ms. EF, Ve and Vp are all ratios and therefore are unit-less, ± indicates plus and minus around one standard deviation (SD)

*Ang1 and 2* angiopoietin 1 and 2, *FGFb* fibroblast growth factor beta, *HGF* hepatocyte growth factor, *IL6 and 8* interleukins 6 and 8, *KGF* keratinocyte growth factor, *CK18* cytokeratin 18, *PDGFbb* platelet-derived growth factor bb isoform, *PlGF* placental growth factor, *SDF1b* stromal-derived growth factor beta, *VCAM1* vascular cell adhesion molecule 1, *VEGFA, C, D, R1, and R2* vascular endothelial growth factor A, C, and D and receptors 1 and 2, *ADC* apparent diffusion coefficient, *EF* ejection fraction, *ETV* enhancing tumor volume, *WTV* whole tumor volume, *IAUC* initial area under the contrast agent concentration curve, *Ktrans* endothelial transfer constant, *Ve* extracellular extravascular space fractional volume, *Vp* plasma fractional volume

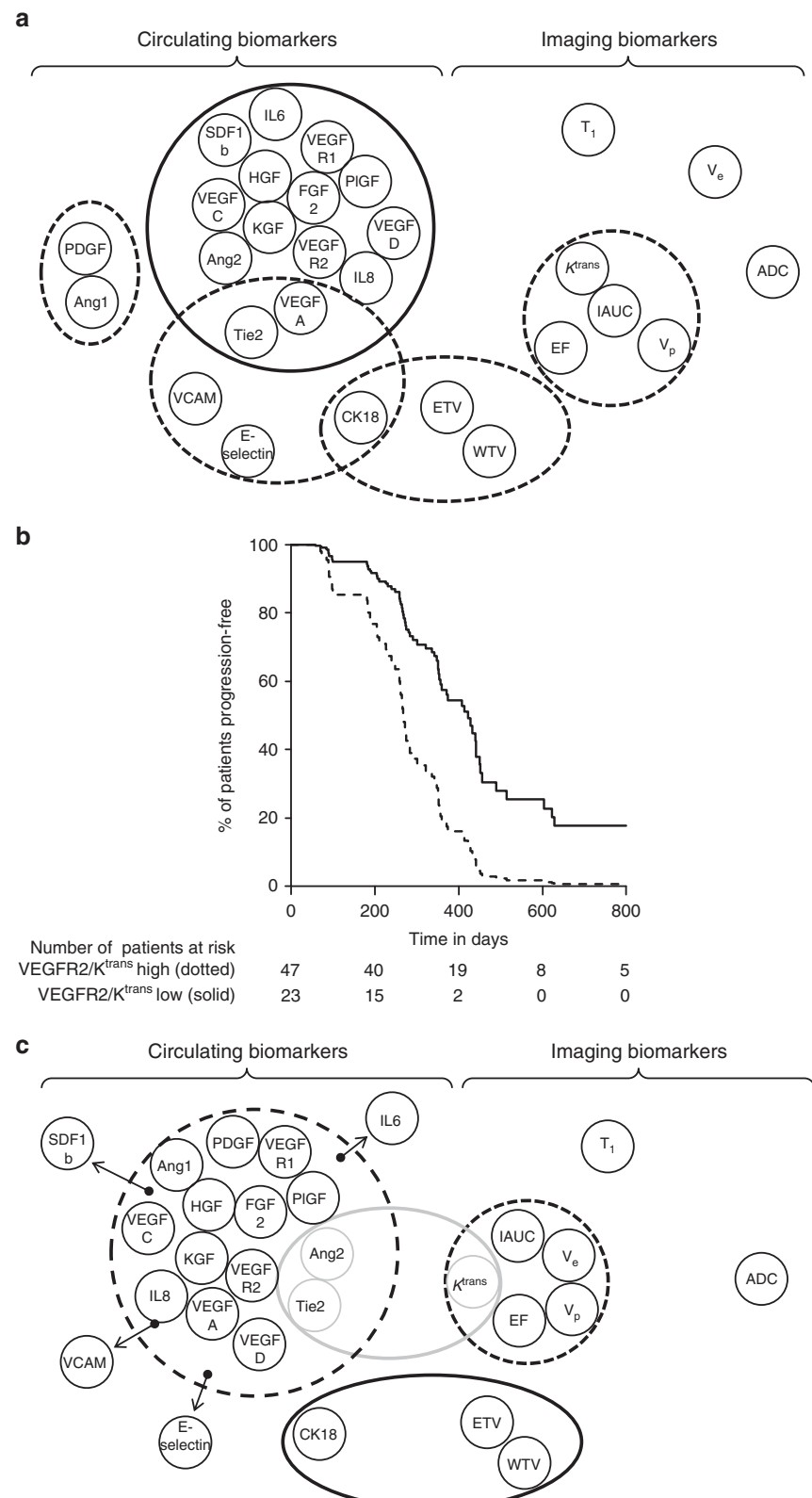

**Bevacizumab induces correlation of Ang2, Tie2, and $K^{trans}$.** Following treatment with single agent bevacizumab, the strong correlations seen between many circulating biomarkers were weakened (median reduction of $r = 0.15$, $r$ ranges from $-0.22$ to $0.70$). This disruption of the core network relationships of circulating biomarkers (Fig. 2c) implied that bevacizumab impacted

specific aspects of the tumor neo-vasculature. The relationships of imaging biomarkers to one another were essentially unchanged during this period (Fig. 2c).

The most significant change observed was development of a correlative relationship between Ang2, Tie2, and tumor $K^{trans}$, where the correlation between circulating Tie2 and tumor $K^{trans}$

**Fig. 2** Correlation networks of circulating and imaging biomarkers. **a** Pearson's correlation networks were constructed for circulating and imaging biomarkers measured at baseline. Clusters with median correlation coefficients above 0.5 are shown in thick black lines, while <0.5 but ≥ 0.35 are shown with dotted lines and correlations <0.35 are not displayed. **b** Compares PFS outcomes in two groups of patients defined by the ratio of pre-treatment VEGFR2:Ktrans, where the cut-off was selected as the 33rd percentile of the ratios. Thus 24 patients were included in the worse prognostic group (dashed black line) and 46 in the better prognostic group (solid black line). The median survival intervals of the two groups were 248 (range 58–423) and 348 (73–1750) days ($p = 0.0008$, Log rank test), respectively. Prognostic factors such as performance status and tumor volume were controlled for by using Cox proportional hazard analysis. The hazard ratio demonstrated that patients with high VEGFR2:Ktrans ratio have a significantly greater risk of progression than the other cohorts (HR 3.01, $p = 0.00014$, Wald test). This defines this cohort as the poor prognostic group, we compared the group's WTV derived from DCE-MRI data with that of the other patient groups. **c** Pearson's correlation networks were constructed for circulating and imaging biomarkers measured after two weeks of treatment with bevacizumab; tight correlations between circulating biomarkers were lost and some biomarkers were completely removed from the correlative relationship (arrows). In contrast bevacizumab induced a correlative relationship between Ktrans, Ang2, and Tie2 (gray line). Clusters with median correlation coefficients above 0.5 are shown in thick black lines, while <0.5 but ≥ 0.35 are shown with dotted lines and correlations <0.35 are not displayed. Ang1 and 2, angiopoietin 1 and 2; FGFb, fibroblast growth factor beta; HGF, hepatocyte growth factor, IL6 and 8, interleukins 6 and 8; KGF; keratinocyte growth factor, CK18; cytokeratin 18, PDGFbb; platelet-derived growth factor bb isoform, PlGF; placental growth factor, SDF1b; stromal-derived growth factor beta, VCAM1; vascular cell adhesion molecule 1; VEGFA, C, D, R1 and R2; vascular endothelial growth factor A, C and D and receptors 1 and 2, ADC; apparent diffusion coefficient, EF; ejection fraction, ETV; enhancing tumor volume, WTV; whole tumor volume, IAUC; initial area under the contrast agent concentration curve; $K^{trans}$; endothelial transfer constant, Ve; extracellular extravascular space fractional volume, Vp; plasma fractional volume, DCE-MRI; dynamic contrast-enhanced magnetic resonance imaging

**Table 3 Univariate analysis of biomarkers before treatment**

| Parameter | *p*-value |
| --- | --- |
| Total tumor volume[a] | 0.0001 |
| WVT[b] | 0.004 |
| EVT[b] | 0.004 |
| EF | 0.004 |
| LDH | 0.005 |
| ECOG performance status | 0.006 |
| VEGFA | 0.008 |
| IAUC | 0.012 |
| $K^{trans}$ | 0.017 |
| Cytokeratin18 | 0.13 |
| VEGFR2 | 0.19 |

Clinical, circulating and imaging biomarkers were analyzed using univariate Cox proportional hazard regression analysis to examine their prognostic significance with respect to progression-free survival. Biomarkers with univariate *p*-values that were less than 0.05 (listed in this table) were included in the subsequent multi-variate analysis (Table 4)
[a]Total tumor volume calculated from CT scan
[b]WTV and ETV calculated from MRI scan
*EF* ejection fraction, *ETV* enhancing tumor volume, *WTV* whole tumor volume, *LDH* lactose dehydrogenase, *ECOG* Eastern Cooperative Group, *VEGFA and R2* vascular endothelial growth factor A and receptor 2, *IAUC* initial area under the contrast agent concentration curve; $K^{trans}$ endothelial transfer constant, *MRI* magnetic resonance imaging

emerged and increased from −0.21 pre-treatment to 0.47 following treatment. This finding suggested that Tie2 was a biomarker closely related to tumor vasculature. The correlation between Tie2 and Ang2 was also strengthened, corroborating our previous findings of increased bevacizumab-induced correlation in the Ang/Tie pathway, in ovarian cancer[12].

Finally, we evaluated the effect of subsequent addition of cytotoxic chemotherapy to bevacizumab on the network of biomarkers of the neo-vasculature. Cytotoxic chemotherapy did not significantly impact further on any correlations, within the circulating biomarkers alone, within the imaging biomarkers alone, or between the circulating and imaging biomarkers (Supplementary Figure 4).

Our attention had already been directed towards Tie2 as we had previously reported in ovarian cancer that bevacizumab induced a correlation between Angiopoietins (Ang) and Tie2 and that Tie2 was of clinical significance in the disease[11,12]. The data presented here validate this finding in colorectal cancer, where the bevacizumab-associated biomarker signature not only includes Ang2 and Tie2 but also importantly revealed that bevacizumab induced a correlative relationship between Tie2 and the tumor

vascular imaging parameter $K^{trans}$ (Fig. 2c). Taken with other reports of PD changes in Tie2 induced by VEGFi, the implication is that Tie2 is a pan-tumor PD biomarker for antibody and small molecule VEGF inhibitors. Further, through analysis of the pre-treatment values of biomarkers (Table 2) and their changes during treatment (Supplementary Figure 1a–c), we have here defined a bevacizumab-induced vascular response as a greater than 30% reduction in Tie2 concentration since this exceeded the 95% confidence intervals of treatment-free variations in Tie2.

**The Tie2 signature is derived from tumor vasculature.** Correlative studies, performed to avoid the confounder of multiple comparisons, revealed in ovarian cancer[12] and in this study in colorectal cancer an induced relationship between Tie2 and its ligands in patients' plasma samples. Here, in colorectal cancer, network (Fig. 2) and biomarker data (Supplementary Figure 1 and Table 2) revealed that bevacizumab induced correlations between Ang2, Tie2, and $K^{trans}$, the imaging parameter most frequently employed in studies of VEGF inhibitors[15], which is a composite of endothelial surface area and vascular permeability. Since the Ang2–Tie2 pathway has been implicated in vessel destabilization[16] the reductions in value and coordination of these parameters observed here potentially reflect vascular normalization[17], can be detected through changes in $K^{trans}$[8].

We extended the evaluation from the correlative two time point analysis presented above to a more extensive and dynamic analysis of Tie2, tumor $K^{trans}$ and the epithelial biomarker, CK18. Here we used an unsupervised hierarchical clustering analysis to avoid arbitrary classification of patients into early and late responders. This analysis separated patients into two distinct cohorts with different Tie2 trajectories (Fig. 3a, $p = 1.8 \times 10^{-6}$, Mann–Whitney U test). Taking the same cluster-defined populations, we plotted the trajectories of tumor $K^{trans}$ (Fig. 3b, $p = 0.003$, Mann–Whitney U test) and CK18 (Fig. 3c, $p = 0.18$, Mann–Whitney U test) showing that the cluster-defined behavior of Tie2 (Fig. 3a) closely resembled tumor $K^{trans}$ (Fig. 3b) but not that of CK18 (Fig. 3c). Thus, as bevacizumab induced a correlation between $K^{trans}$ and Tie2 shortly after treatment started and the trajectories of the two Tie2 clusters mirrored changes in tumor $K^{trans}$ but not CK18, the data imply that Tie2 reflects bevacizumab-induced tumor vascular modulation. This is important as our data[11,12] and the PD data of others[2,3,6] had suggested that the Tie2 change was of vascular origin. Here, the imaging data imply that the Tie2 signal originates from tumor vasculature or at least through a tumor vasculature-associated

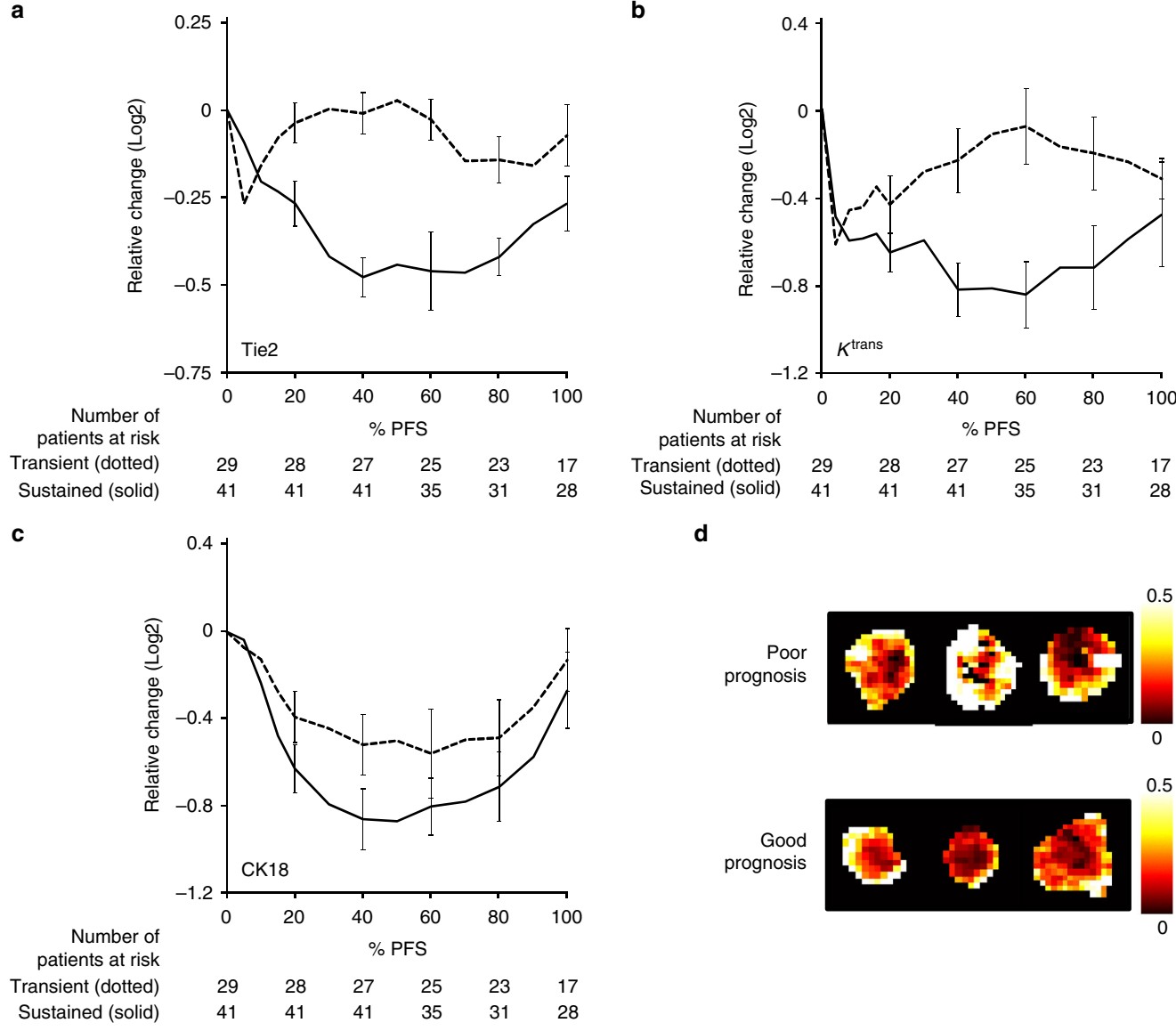

**Fig. 3** Changes in Tie2 but not CK18 reflect tumor vascular control ($K^{trans}$). Two cohorts (or clusters) of patients were identified through an unsupervised hierarchical clustering of Tie2 trajectories (**a**). In one cohort, there was an immediate but transient reduction (dotted line) in Tie2, while in the other cohort there was a later but more sustained reduction (solid line). The two cohorts behaved significantly differently ($P = 1.8 \times 10^{-6}$, Mann–Whitney $U$ test). Similar cohorts of patients were identified with $K^{trans}$ values (**b**), which again behaved significantly differently ($P = 0.003$, Mann–Whitney $U$ test). The pattern of behavior of CK18 was different (**c**), and the two cohorts were not significantly different ($P = 0.18$, Mann–Whitney $U$ test). This indicates that Tie2 reflects the impact of bevacizumab on tumor vasculature. Error bars indicate the standard error. **d** Shows an example of $K^{trans}$ parameter maps, which reflect the tumor behavior of the two cohorts, shown in **b**. The largest difference in parameter maps can be seen in the two middle position maps, which correspond to the maximum difference between the two cluster-derived curves. CK18; cytokeratin 18, $K^{trans}$; endothelial transfer constant

cell lineage. This finding is consistent with the known data documenting expression of Tie2 in vessels[18] and its modulation by VEGF[4,5].

**Prolonged reductions in Tie2 are associated with better PFS.** Our data[11,12] and those of others[2,3,6] demonstrate that Tie2 is a pan-tumor, vascular PD biomarker for VEGF inhibitors. As the Ang2–Tie2 pathway is heavily implicated in mediating[16,18,19] angiogenesis, we hypothesized that those patients who had the greatest and most prolonged VEGFi-induced reductions in Tie2 should have fared the best and vice versa.

As a first step in the analysis of the prognostic significance of Tie2 we tested the association between the two groups of patients defined in the unsupervised clustering analysis of Tie2 (Fig. 3a)

and the pre-treatment clinical and biomarker parameters that had been identified as having independent prognostic significance (the VEGF-R2 to $K^{trans}$ ratio; first section of results). While the Tie2 curves (Fig. 3a) were not associated with pre-treatment clinical prognostic factors, there was a significant association with VEGF-R2 to $K^{trans}$ ratio ($p = 0.014$, Fisher's exact test, Supplementary Table 1) in which patients with the higher ratio manifested less reduction in Tie2. Thus, there is a group of patients, who are defined by a high VEGF-R2/$K^{trans}$ ratio and less significant and durable reductions in Tie2, who have a shorter PFS (Fig. 2b: median PFS 248 vs. 348 days, $p = 0.0008$, Log rank test. HR for progression: 3.01, $p = 0.00014$, Wald test. As VEGF-R2 is most likely derived from the endothelial bed reflecting the volume of tumor microvasculature and tumor $K^{trans}$ and the

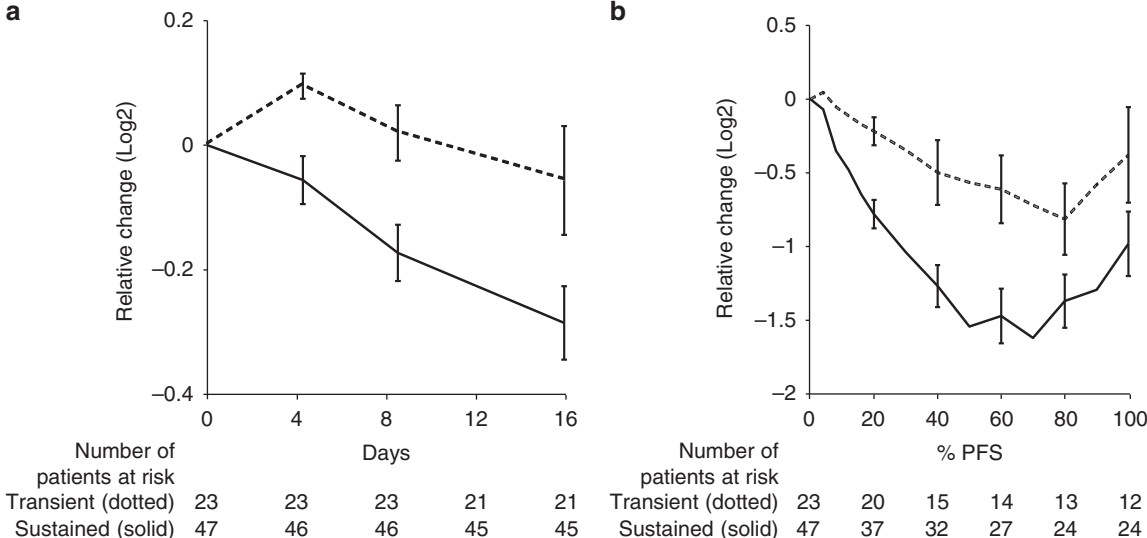

**Fig. 4** Prognostic value of angiogenic biomarkers. The association between a less profound reduction in Tie2 and higher $K^{trans}$:VEGFR2 ratio (Fig. 2b) identified a poor prognostic group. The WTV derived from DCI-MRI of this poor prognostic group (dotted lines) was compared with better prognostic outcome group (solid lines). **a** Shows the differential impact of single agent bevacizumab (10 mg/kg) on WTV between the two groups up to 15 days after treatment, with effects emerging by day 4 ($P = 0.029$, Mann–Whitney $U$ test). **b** Volumetric analysis showed persistence of this difference throughout the course of combined cytotoxic chemotherapy and bevacizumab, with changes in WTV plotted using the same groups against the percentage time that elapsed between randomization and the date of progression/censoring (%PFS; $P = 0.012$, Mann–Whitney $U$ test). Error bars indicate the standard error. WTV; whole tumor volume, DCE-MRI; dynamic contrast-enhanced magnetic resonance imaging, VEGFR2; vascular endothelial growth factor receptor 2, PFS; progression-free survival

Ang/Tie2 axis are biomarkers that reflect the sensitivity of the tumor to VEGF inhibitors, the implication is that the worst prognostic group bears tumors that achieve angiogenesis through a non-VEGF dependent pathway.

Having defined better and worse prognostic groups, we evaluated the tumor volumetric response to bevacizumab in each group. WTV did not change in the poor prognosis group but was significantly reduced in the better prognosis patients ($p = 0.029$, Mann–Whitney $U$ test, Fig. 4a). These differences were apparent within 4 days of bevacizumab monotherapy. By the end of a two-week cycle of bevacizumab patients with beneficial outcome had attained a median reduction in WTV to 82% of pre-treatment levels (95% CI: 47–143%). We then analyzed the tumor volumetric change against the percentage time that elapsed between randomization and the date of progression/censoring (%PFS). This analysis demonstrated that a significant difference in tumor response remained throughout the clinical course of combination therapy with bevacizumab and cytotoxic chemotherapy ($p = 0.012$, Mann–Whitney $U$ test, Fig. 4b).

Taking these data together, we have identified a group of patients with significantly worse PFS. The patients' tumors were characterized by high VEGF-R2 to $K^{trans}$ ratio before treatment, by a lack of tumor size reduction after two weeks' single dose of bevacizumab and by transient bevacizumab-induced vascular control that was reflected by less profound and less durable reductions in Tie2. The data imply that patients in the poor prognosis group had tumors that were less dependent on VEGF signaling.

**Validation of the multi-tissue compartment model of cancer.** Our previous ovarian cancer studies[11,12] provided the biomarker data to suggest that the vascular and epithelial compartments were additive targets for anti-cancer treatment. Therefore, we investigated the colorectal cancer data for the same relationship as, in conjunction with clinical trial data[7,20–24], such a model would suggest that the tumor vascular and epithelial tissue compartments should be targeted at each clinical recurrence of angio-sensitive cancers that are usually treated with cytotoxic chemotherapy and VEGFi.

The data showed that bevacizumab imposed an association between Tie2, Ang2, and tumor $K^{trans}$ and that the impact on Tie2 reflected vascular changes in tumor, volumetric response, and PFS. We then determined whether changes in the epithelial and vascular compartments were independent of each other. The analysis is based on the identification of an inflection point in biomarker data where, for instance, a parameter such as Tie2 has been decreasing over time and then starts to increase. This Bayesian hierarchical modeling approach allows evaluation of different mathematical rules for the calculation of the inflection point, which is considered to infer a change in biology. In the case of Tie2 an inflection point suggests vascular progression, whereas in the case of CK18 an inflection point suggests epithelial progression. The approach also generates putative rules that could subsequently be used for clinical decision making.

Based on these biomarker-defined response and progression rules, tumor progression predicted by circulating CK18 and Tie2 was summarized to identify the behavior of the level of the entire cohort. Critically, the performance of predictions made from actual data derived from the two biomarkers (66%) was superior to the performance of either biomarker alone; 54 and 41%, respectively Fig. 5a).

In a complex clinical dataset there is often some missing data. The Bayesian hierarchical modeling approach allows interpolation to simulate missing data and provide a "full" dataset. Simulated data demonstrated that the combined biomarkers were capable of predicting progression in 77% of patients, which exceeded the result from actual data that had some missing data points. Once again, this finding showed that prediction of outcome from combined Tie2 and CK18 biomarkers provided better performance compared to either biomarker alone ($p < 0.0001$, Mann–Whitney $U$ test). Further, we found evidence of temporal disconnect between the timing of progression in the

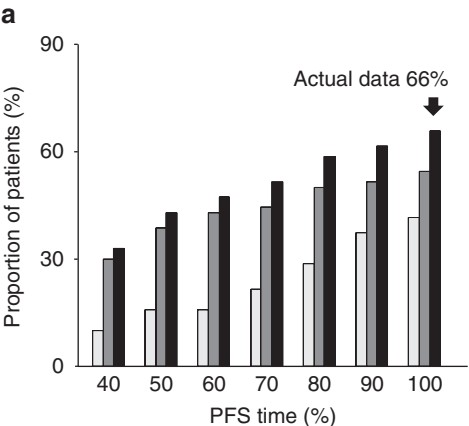
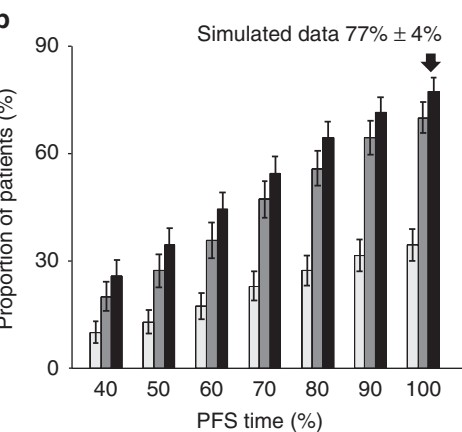

**Fig. 5** Predicting disease progression using plasma CK18 and Tie2 levels. The figure shows the potential of Tie2, cytokeratin 18 (CK18), and both biomarkers modeled together to predict disease progression. The prediction made by CK18 alone is shown in light gray, by Tie2 alone is shown in mid-gray and using the combination of CK18 and Tie2 in black. **a** Shows predictions based on actual data where the combined biomarkers predicted progression in 66% of patients, whereas single biomarker predictions of progressive disease were achieved in 54 and 41%. Interestingly the prediction derived from CK18 occurred later than that from Tie2 with a lead-time of 50 days (14% PFS) as the increase in the Tie2 biomarker predates that of CK18. **b** Shows the same calculation but using simulations to fill any missing data. Again a significant improvement on the number of predictions was observed when Tie2 and CK18 were used jointly for prediction ($p < 0.0001$, Mann–Whitney $U$ compared with either single antigen), revealing clear evidence of additivity between the two biomarkers with respect to predicting progressive disease. Error bars indicate the standard error. CK18; cytokeratin 18, PFS; progression-free survival

vascular and the epithelial compartments. Across the entire cohort, Tie2 predicted progression before CK18 with an average lead-time of 50 days ($p = 0.026$, Mann–Whitney $U$ test).

In all, these data suggest that RECIST-defined PD can best be anticipated through a combination of multimodal biomarkers that reflect both vascular progression and epithelial progression. Thus, epithelial and vascular biomarker data improve the modeling of progressive disease when considered together, supporting the hypothesis that the two tissue compartments may be valid targets at treatment for initial presentation and re-treatment in patients with disease relapse (Fig. 6a).

**Using Tie2 to define vascular response and progression**. Estimates of pre-treatment intra-patient variation indicated that the 95% CI of Tie2 and CK18 were 0.53 and 0.42 respectively (log2 scale, Supplementary Figure 3). Thus, a Tie2-defined vascular response can be inferred when its plasma concentration is reduced by at least 30% during the first 3 cycles of treatment. A CK18 reported epithelial response can be inferred when its plasma concentration is reduced by at least 25% over this time frame. The Bayesian hierarchical modeling approach revealed that a 50 and 40% increase above the nadir concentrations of plasma CK18 and Tie2, respectively, were indicative of disease progression (Fig. 6b–e; Supplementary Figure 5).

Looking at the individual patient data (Fig. 6b–e), it becomes clear that some patients develop tissue compartment progression in vascular, epithelial or both compartments by the point of conventional RECIST-defined disease progression. The advance that Tie2 brings is that a VEGFi can be stopped much earlier than conventional management would dictate, if there is no Tie2 defined response or if Tie2 reported progression occurs. On the other hand an epithelial biomarker progression defined through increases in CK18 can sometimes precede Tie2-defined vascular biomarker progression. Such patients may benefit from continued VEGFi beyond RECIST progression as they may experience significantly decelerated tumor growth that persists until Tie2-defined vascular biomarker progression occurs.

## Discussion

Anti-angiogenic VEGFi are widely prescribed in oncology yet their effective use has been compromised by the lack of

biomarkers to optimize their clinical utility. Here, we have studied a panel of 19 circulating and 9 imaging biomarkers in 70 patients with metastatic colorectal cancer during treatment with bevacizumab followed by cytotoxic chemotherapy and bevacizumab. The data were analyzed using an unsupervised statistical approach to avoid the confounder of multiple comparisons and to avoid arbitrary classification of biomarker patterns. While changes in other biomarkers occurred and are of potential interest, our attention was drawn to Tie2 because single agent bevacizumab induced a correlation between Ang2, Tie2, and $K^{trans}$, whereas the relationship between several other biomarkers was unchanged or weakened.

This approach allowed us to demonstrate that Tie2 is a tumor vascular response biomarker for bevacizumab in patients with metastatic colorectal cancer. The imaging data demonstrate that in solid tumors the Tie2 biomarker signal is derived from tumor vasculature and that it is of clinical significance. As we have now demonstrated the value of circulating Tie2 in ovarian[11,12] and colorectal cancers, when treating patients with bevacizumab-cytotoxic chemotherapy combinations and others have described similar findings in colorectal cancer[3], glioma[2] and gall bladder[6] cancer in trials of the VEGF RTKi, cediranib, it is credible that Tie2 is a generic circulating tumor vascular response biomarker for VEGF inhibitors. Our confidence in Tie2 as a generic response biomarker for VEGF inhibitors is thereby enhanced by these findings in four tumors types, using two different classes of VEGF inhibitor in at least two different laboratories[2,3].

Our findings also provide some key data to support clinical translation of the Tie2 assay. The small intra-patient variation in pre-treatment concentrations of Tie2 provides evidence of assay precision increasing confidence in the potential clinical utility of this biomarker. Further, we have shown that a vascular response can be defined as a confirmed reduction in Tie2 concentrations of more than 30% while the biochemical definition of vascular progression is a greater than 40% increase in plasma Tie2 concentrations above the nadir. The suitability of these criteria to inform clinical decisions can now be tested formally in independent datasets to help develop guidelines for interpreting a clinically approved Tie2 assay.

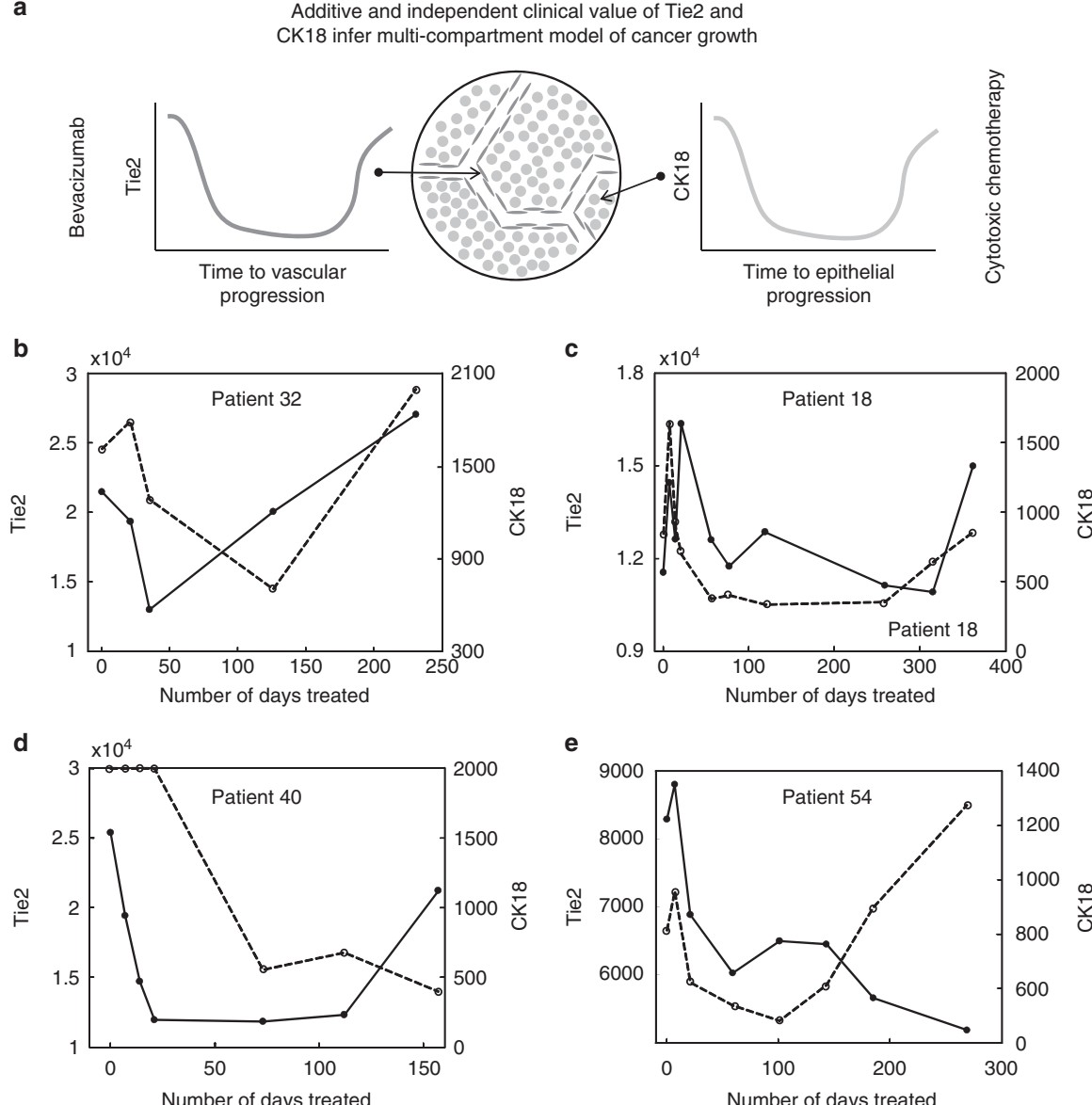

**Fig. 6** The tissue multi-compartment model of cancer treatment. **a** The biomarker data show that Tie2 is a tumor response biomarker for VEGFi and that the predictive value of vascular biomarker data adds to epithelial biomarker data to improve modeling of progressive disease. Together with our previous data in ovarian cancer, the implication of these findings is that the vascular and epithelial compartments are distinct, valid and useful concepts in the treatment of each recurrence of solid tumors. Examples of individual patient data are shown in **b**–**e**; the x-axis represents the number of days that the patient received treatment until they developed RECIST-defined progression (i.e., total tumor burden), Tie2 data are shown as solid black lines and CK18 in dotted black lines. The units for both biomarkers are pg/ml. In **b** Patient 32 has undergone an epithelial response, vascular response, epithelial progression, and vascular progression. In **c**, the patient had an epithelial response but not a vascular response thus the patient could have stopped bevacizumab after ~8 weeks of treatment. The patient subsequently had epithelial and vascular progression. In **d** the patient had an epithelial and vascular biomarker response and then developed vascular progression but not epithelial biomarker progression. In **e** the patient attained an epithelial and vascular response with subsequent epithelial progression but not vascular progression, suggesting that the vasculature was still controlled at the point of RECIST progression. CK18; cytokeratin 18, VEGFi; vascular endothelial growth factor inhibitor, RECIST; response evaluation criteria in solid tumors

Accepting the clinical value of Tie2, the next question is whether Tie2-defined vascular progression reflects a functional role of Ang2/Tie2 in mediating resistance to VEGF inhibitors[4]. This is critical as, while we know that single agent Ang/Tie2 inhibitors are clinically active[8], most phase III trials of Ang/Tie2 inhibitors have not selected patients on the basis of biomarkers and the results to date have only been modestly positive[10]. Biomarker-directed trials could test this hypothesis. Several mechanisms of resistance to VEGFi have been described[25] and studies have suggested that

acquired resistance to VEGF inhibitors might be mediated by macrophages[26–28]. Thus an alternative hypothesis about the putative role of the Ang-Tie2 axis in effecting resistance to VEGFi, is that acquired resistance to VEGFi is mediated through Tie2-expressing or -modulated monocytes/macrophages. This alternative hypothesis is consistent with our observed correlation between Ang2 and Tie2, which might reflect local Ang2-mediated recruitment of Tie2-expressing macrophages at the point of vascular progression[29]. Further, a recent publication has shown that

macrophage-mediated resistance to VEGFi could be reversed by administration of inhibitors of macrophage CSF-1 receptors[30], in vivo, potentially highlighting a strategy to overcome resistance to VEGFi, when Tie2 increases. To some extent questions over the functional roles of different cell types in mediating VEGFi resistance will only be answered through detailed and warranted studies of serial tumor biopsies.

Our study, taken in conjunction with others cited above, suggests that Tie2 is a generic VEGFi response biomarker, but some caveats should be acknowledged. The mechanisms responsible for the changes in Tie2 have not been defined at the microscopic level although this should not detract from the potential clinical utility of the biomarker in view of the macroscopic evidence derived from advanced imaging. To date only a few hundred patients have provided samples for these assays and these patients have all been recruited from within clinical trials that evaluated multiple potential biomarkers simultaneously. Real world evaluation in larger populations is now required to define the clinical value of the assay. Further, while we have defined a vascular response as a greater than 30% reduction in Tie2, the duration over which this should be studied is not clear although existing data suggest that such an effect should be seen within 9 weeks of starting treatment.

Randomized trials in the first-line, second-line, and third-line colorectal/ platinum-resistant ovarian cancer settings[7,20–24,31] show that the addition of VEGF pathway inhibitors to cytotoxic therapy improves PFS[1]. The question is how to use the information derived here to improve use of VEGF inhibitors and thereby translate these improvements in PFS into improved OS. In this study we have demonstrated the independent and additive value of modeling both epithelial and vascular compartments in cancer patients. The clinical trial results and our now validated model emphasize the critical importance of considering the vasculature as a target for re-treatment with anti-angiogenic agents in patients previously treated with such drugs. Rather than restricting anti-angiogenic agents to initial presentation, these agents should be instituted and maintained in angio-sensitive tumors until Tie2-defined vascular progression at each episode of progressive disease.

Extrapolating further, a "multi-compartment" model of cancer care may in due course encompass the many other phenotypic compartments that represent the "hallmarks of cancer", which were described by Hanahan and Weinberg[32], including activation of invasion and metastasis and avoidance of immune destruction. Thus, in the same way that our data suggest that use and re-use of VEGFi should be considered at each recurrence of angio-sensitive cancer using Tie2 to guide therapy, so in time the same principles might also apply to immunotherapies throughout patients' lives with cancer.

## Methods

**Clinical trial**. This study was granted approval from the National Research Ethics Service, Central Manchester Research Ethics Committee on 15 June 2009 (REC Reference 09/H1008/99). The study was registered with EudraCT (number 2009-011377-33). The study was performed in adherence with the relevant ethical guidelines and all patients granted written informed consent prior to enrollment in this study. In this prospective trial seventy patients with a new diagnosis of metastatic colorectal cancer, whose tumors were amenable to advanced imaging, were treated with bevacizumab 10 mg/kg to define a drug specific biomarker signature followed, two weeks later, by FOLFOX-6 or XELOX with bevacizumab at a dose intensity of 2.5 mg/kg/week until progressive disease occurred; the primary endpoint. All patients received bevacizumab at a dose intensity of 2.5 mg/kg/week along with combined oxaliplatin and a fluoropyrimidine (5-fluorouracil, 5FU), a version of FOLFOX-6 based on the NO16966 trial regimen[33]. Briefly, for patients with a central venous catheter, oxaliplatin 85 mg/m$^2$, folinic acid 175 mg, and bevacizumab 5 mg/kg were administered as short infusions. Then bolus 5FU 400 mg/m$^2$ was administered and followed by 5FU 2400 mg/m$^2$ over 46 h. This was administered on a 2-weekly schedule. For patients who did not have a central venous catheter, oxaliplatin

130 mg/m$^2$, and bevacizumab 7.5 mg/kg were administered intravenously every 3 weeks with oral capecitabine 1000 mg/m$^2$ bd for 14 of every 21 days. Dose modifications were made in accordance with local protocols for Common Toxicity Criteria for Adverse Events (CTC AE) grade 3 or 4 toxicities. Tumors were radiologically evaluated every 12 weeks according to RECIST 1.0. If oxaliplatin was withdrawn for toxicity, patients could continue treatment with 5FU and bevacizumab. Patients were recruited between 2009 and 2012 and followed up until 2016.

Eligible patients had a performance status of 0–2, were aged 18 years or more, and had a life expectancy of 12 or more weeks. Adequate bone marrow, hepatic and renal function including proteinuria <2+ on urine dipstick, were required. The INR and APTT were a maximum of 1.5 fold the upper limit of normal. Eligible patients had to have disease that was measureable (3 cm or more) and amenable to diffusion-weighted MRI (DW-MRI) and dynamic contrast-enhanced magnetic resonance imaging (DCE-MRI). Imaging focused on liver metastases that were between 3 and 10 cm diameter. Exclusion criteria included previous treatment for metastatic disease (adjuvant therapy at least 12 months earlier was permitted), brain metastases, spinal cord compression, uncontrolled inter-current illness, pregnant or breast-feeding women and surgery, significant injury or radiotherapy in the 4 weeks prior to first treatment. Previous treatment with VEGF pathway inhibitors was not permitted and hypertensive patients were excluded. Patients with clinically significant cardiovascular disease or thrombo-embolism within the previous 6 months were excluded. Patients with hemorrhagic diathesis, those receiving aspirin at 325 mg/day or more and those who had started therapeutic anti-coagulation within the previous 4 weeks were ineligible. Additional exclusion criteria included another malignancy within the previous 5 years, dihydro-pyrimidine dehydrogenase deficiency, non-healing wound/fracture, pre-existing grade 2 neuropathy, and uncontrolled bowel disorders.

The study (EudraCT 2009-011377-33) was sponsored by The Christie NHS Trust and ethical approval was given by the NHS Health Research Authority National Research Ethics Service, North West Greater Manchester Central ethics committee (reference 09/H1008/99). All patients provided written informed consent.

**Biomarkers**. Patients underwent two pre-treatment scans, for DW-MRI and DCE-MRI within the two weeks before the first administration of bevacizumab. At the same time, two 6 ml blood samples were drawn to quantify pre-treatment plasma concentrations of circulating biomarkers. These samples were used to establish the repeatability of imaging and circulating biomarker data and to generate the 95% confidence intervals that would define whether treatment induced a statistically significant effect on each biomarker[34].

A first dose of bevacizumab 10 mg/kg was administered on day 1 and further imaging and circulating biomarker samples were collected on days 3, 8 (imaging only), and 15, as guided by our previous work[35] (Table S2). On day 15, after biomarker collection, the oxaliplatin-fluoropyrimidine cytotoxic therapy was initiated and further imaging and circulating biomarkers were collected on day 22 (7 days into the first cycle of bevacizumab combined cytotoxic therapy), at week 6, at 6 months and at disease progression.

All biomarker studies were carried out by staff that were blinded to the clinical demographic and outcome data. DW-MRI and DCE-MRI were performed and plasma samples aspirated before and during treatment up to the point that progressive disease was diagnosed (Table S1). The biomarker schedule and methods are described in Supplementary Methods 1.

**Circulating biomarkers**. Blood samples for measurement of angiogenesis associated analytes were drawn directly into 6-ml ethylenediaminetetraacetic acid (EDTA) vacutainers and plasma was separated by centrifugation at 3000× *g* for 10 minutes. Plasma was immediately aspirated, separated into aliquots and frozen at a maximum of −20 °C immediately and −80 °C in the laboratory. Samples were only thawed once for each analysis, which was performed in triplicate. Enzyme-Linked Immuno-Sorbent Assays (ELISAs) were performed using Search-Light chemiluminescent arrays and SearchLight Plus charged couple device imaging system (Aushon Biosystems, Boston, US). These multiplex assays allowed quantification of up to six different proteins in each well of a ninety-six well plate. All assays were performed in the Clinical and Experimental Pharmacology Group laboratories, Cancer Research UK Manchester Institute to Good Clinical Practice (GCP) standard and underwent in-house validation as previously described[36]. Protein concentrations were assessed using six-plex ELISAs of Ang2, FGFb, HGF, PDGFbb, VEGF-A and VEGF-C and IL6, IL8, KGF, PlGF, VEGF-R1, VEGF-R2; duplexes of Ang1 and Tie2 and E-selectin and VCAM-1 and single plexes of VEGFD and SDF1β. The M65 ELISA for cytokeratin 18, (Peviva, Bromma, Sweden), was used to quantify total epithelial cell death[37]shown previously to be a biomarker of tumor burden in colorectal cancer[13]. The biomarkers generally follow log-normal distribution (Supplementary Figure 1). Therefore their longitudinal changes were investigated in log2 scale to reduce biomarker variation inherent in patients. Technical variations and pre-treatment intra-patient variation of these proteins were estimated based on log2 transformed ratios of two technical replicate measurements or two consecutive pre-treatment measurements.

**Imaging biomarker acquisition and analysis**. DW-MRI and DCE-MRI were performed in one scanning session. All imaging acquisition and analysis was performed in the Quantitative Biomedical Imaging laboratory, University of Manchester, to Good Clinical Practice (GCP) standards. Data were acquired on a 1.5 T Philips Achieva scanner (Philips Healthcare, Best, The Netherlands). DW-MRI images were acquired using a non-breath holding, fat-suppressed, spin echo, echo planar imaging sequence with the following parameters: FOV 375 mm × 375 mm; slice thickness 4 mm; number of slices 25; matrix 256 × 256; in-plane resolution 1.46 mm × 1.46 mm; TR 3416 ms; TE 90 ms with diffusion sensitization ($b$ values) of 0, 150, 500 and 800 s/mm$^2$. MRI protocol was carried out as previously described and included a turbo spin-echo $T_1$-weighted sequence; $T_2$-weighted turbo spin-echo sequence and a 3-D DCE-MRI protocol using an RF-spoiled fast field echo ($T_1$-FFE) sequence[38]. During the dynamic sequence, 0.1 mmol/kg of gadoterate meglumine contrast agent (Dotarem, Guebert, France) was administered intravenously at the sixth dynamic time-point at a rate of 3 ml/s, using a Medrad Spectris power injector (Bayer Healthcare Pharmaceuticals, USA).

DCE-MRI data were analyzed using in-house software, Manchester Dynamic Modeling (MaDyM, Manchester, UK). Regions of interest (ROIs) were defined manually for the whole tumor volume using Java Image software (JIM version 5.0, Xinpase Systems Ltd, UK) and with reference to the $T_1$- and $T_2$-weighted images, as well as the DCE-MRI images. The arterial input function (AIF) was determined for each patient visit using an automated technique applied to the nearest feeding artery[39] and corrected for patient hematocrit. The individual patient pre-treatment hematocrit readings were used. Parameters that were reported included whole tumor volume, enhancing volume, enhancing fraction, tumor median values of the initial area under the contrast agent concentration curve over the first 60 seconds (IAUC$_{60}$[40]), and the endothelial transfer coefficient ($K^{trans}$), plasma fractional volume ($v_p$), and extracellular extravascular space fractional volume ($v_e$), as derived using the extended Kety model[41], and tumor pre-contrast longitudinal relaxation time $T_1$. DW-MRI was analyzed by fitting the data to obtain the water apparent diffusion coefficient (ADC). Voxel-wise estimates of ADC in each ROI were calculated by voxel-wise fitting of the diffusion weighted image data using $S(b) = S_0 \times e^{-b \times ADC}$. ADC maps were generated and median tumor ADC was calculated to summarize each tumor.

**Statistical design and analysis**. The study plan was to recruit 70 evaluable patients on the grounds that, based on data accrued a year after the 70th and last patient had entered the study (90% event rate), analysis of the biomarker data would have an 80% power to detect biomarker stratified groups with a hazard ratio of at least 2 in progression free survival, using a Cox proportional hazard regression analysis at a 5% significance level.

Network analysis of all biomarkers was performed before treatment, during the first two weeks and up to 6 months of combination therapy to identify drug induced changes in correlations between biomarkers ("qgraph" package[42] in R 3.2[43]), while avoiding the confound of multiple comparisons. This focused attention on Tie2, the time-courses of which were then clustered and compared with other biomarker data based on the same clusters to determine the origin of the Tie2 signal. The PFS of the two Tie2-defined clusters of patients were compared. Having demonstrated the vascular origin of the Tie2 signal, we examined the prognostic significance of each biomarker in a Cox proportional hazard model with appropriate tests for proportionality and non-linearity. The identified biomarkers defined a cohort of patients with significantly poor prognosis. Tumor volumetric and Tie2 trajectories of these patients during treatment were examined. The trajectories of biomarkers were modeled using Bayesian methods to determine an inflection point where a biomarker's behavior changed during treatment. This was then used to determine when biomarker progression occurred and then to evaluate the additive value of modeling the epithelial and vascular compartments together with respect to progressive disease.

**Network analysis of circulating and imaging biomarkers**. The circulating biomarkers investigated in this study were associated wth angiogenesis biology, epithelial cell death or were conventional clinical tests for tumor burden (platelet count, LDH and CEA). MRI biomarkers were derived from individual tumor lesions as described above. One index lesion was followed per patient. We generated network representations of the correlations between biomarkers, based on: (i) pre-treatment data, (ii) the first two weeks following single agent bevacizumab administration, (iii) and up to six months treatment with bevacizumab and cytotoxic chemotherapy. Matched data enabled correlations between circulating and imaging biomarkers to be evaluated for the pre-treatment data and the data derived from the first two weeks of combination therapy. Changes in circulating and imaging biomarkers were assessed after two weeks of treatment. The network representations minimize the risks of a multiple comparisons analysis, which might result in insufficient statistical power, and allow interpretation based on existing knowledge of disease biology.

**Clustering patients according to changes in Tie2**. The correlation analysis focused attention on Tie2 and we therefore used an unsupervised hierarchical clustering method (based on correlation) to explore maximal separation between Tie2 trajectories. The unsupervised nature of the cluster analysis meant that no predefined thresholds were used. These two groups of Tie2-defined patients were characterized by investigating their (1) association with clinical factors pre-treatment using chi-square test; (2) difference in biomarker levels pre-treatment; (3) difference in trajectories of circulating biomarkers during treatment; and (4) difference in trajectories of imaging and epithelial biomarkers to understand the biological relevance of Tie2.

**Investigating prognostic significance of biomarkers**. The relationships between clinical factors and biomarkers were assessed by correlation (for continuous variables) or chi-squared tests (for categorical variables). The prognostic significance of each candidate biomarker was assessed by including the biomarker either as a continuous variable or through dichotomized analysis based on the distribution median as a sole covariate in a proportional hazards model for progression. In each case, we tested for the corresponding null hypothesis of no effect via a Wald test[44]. Assumption of proportionality was verified based on Schoenfeld residuals[45]. A plot of the Martingale residuals[46] from each marker specific analysis was examined for evidence of nonlinearity in the biomarker–hazard relationship. The covariate was subjected to appropriate transformation/categorization, such as log$_2$ transformation, if the above assumptions were found violated. Biomarkers with univariate $p$-values less than 0.05 were selected for subsequent multivariate analysis, where a backward stepwise method was applied to identify the subset of biomarkers that best explained survival. Interactions of selected biomarkers were explored based on biologically driven hypotheses.

To prevent the estimate from being unduly influenced by a few patients with extreme biomarker measurements, derived models were validated using a bootstrap resampling method. The method makes a more realistic estimation of the distribution of a variable by random sampling of the data with replacements[47]. For each model, 1000 bootstrap samples were generated and the relative frequency of significance for a candidate biomarker was recorded. Biomarkers retained in the optimum model had to have a relative frequency exceeding 66% (significant in at least two cases out of three); a typical threshold used in bootstrap analysis.

The analysis described here were carried out following the REMARK[48] guidelines.

**Clinical significance of patterns in Tie2 trajectories**. The clustering analysis of Tie2 trajectories identified patterns of Tie2 that were associated with worse PFS. Independently Cox proportional hazard analysis identified biomarkers whose pre-treatment values associated with worse PFS. Cut-offs were not pre-defined to avoid data overfitting and multiple comparisons. As a result, the patient cohorts with worse PFS from the two analyses were different. A chi-squared test was applied to investigate whether the two analyses identified a consistent group of patients with worse PFS.

**Dynamics of circulating biomarkers during treatment**. Time-dependent changes in concentrations of each circulating biomarker, measured as log$_2$ ratios to pre-treatment concentrations, were plotted against time elapsed or the percentage time that elapsed between the date of treatment start and the date of progression/ censoring (%PFS; time elapsed divided by PFS). The concept of percentage time was designed to address variation in patient survival[12]. Missing data were interpolated for graphical representation but were not used in any other part of the analysis.

**Modeling biomarker trajectories**. Trajectories of selected biomarkers were modeled using a Bayesian hierarchical modeling approach from actual rather than interpolated data[12]. In brief, changes in biomarker concentrations were approximated to have a piecewise-linear relationship with treatment time where an inflection point separated the decreasing part of the biomarker trajectory from the subsequent increasing component. The inflection point, from a clinical perspective, was hypothesized to reflect a change in tumor behavior and therefore as the earliest sign of tumor progression[12]. This Bayesian modeling approach allowed estimation of the time point at which tumor behavior changes for an individual patient and for the cohort. It also provided natural tolerance to missing data points. This is further described below

**Defining rules for vascular response and progression**. For each biomarker, 95% confidence intervals of its intra-patient variation were calculated by fitting a Normal distribution to the variation, calculated from the two pre-treatment samples. Here, a vascular response was deemed to have occurred when the reduction in the putative response biomarker exceeded the 95% CI within three cycles of treatment. Vascular progression was considered to have occurred if elevation of a

**Table 4 Multivariate analysis of biomarkers before treatment**

|  | Hazard ratio (95% CI) | *p*-value | Bootstrap (frequency of *p* < 0.05) |
| --- | --- | --- | --- |
| Performance status 1 | 1.50 (0.89–2.54) | 0.13 | 35.9% |
| Performance status 2 | 4.96 (1.42-17.41) | 0.012 | 67.3% |
| Total tumor volume[a] | 1.10 (1.05–1.16) | 0.0003 | 79.7% |
| VEGFR2:K$^{trans}$ [b] | 1.01 (1.00–1.02) | 0.0008 | 96.3% |

Biomarkers with univariate *p*-values < 0.05 (Table 3) were included in the subsequent multi-variate analysis. The results from multivariate analysis were validated using a bootstrap resampling method, which examines the stability of the findings through removal of different data points with re-evaluation of the remaining data. In the multivariate analysis all HRs exceeded 1.00 inferring that the higher the parameter/biomarker, the worse the outcome
VEGFA and VEGFR2 were correlated circulating biomarkers but given the relationship between VEGFA and K$^{trans}$, it was biologically more plausible that the signal reflects VEGFR2. To allow comparisons of more than one biomarker with PFS, VEGFR2, and K$^{trans}$ were analyzed as a ratio (VEGFR2:K$^{trans}$)
[a]total tumor volume/100 calculated from CT scan
[b]VEGFR2/Ktrans WTV and ETV calculated from MRI scan
*VEGFA and R2* vascular endothelial growth factor A and receptor 2, *HR* hazard ratio, K$^{trans}$ endothelial transfer constant, *PFS* progression-free survival, *CI* confidence interval, *MRI* magnetic resonance imaging, *CT* computerized tomography scan

biomarker exceeded a designated cut-off from its recorded nadir point during treatment. The optimum cut-off was determined using the developed Bayesian model via inference of pseudo-trials. Full detail on model development and data inference can be found below. The potential of selected biomarkers to predict tumor progression was evaluated based on the rules described above and we then calculated the percentage of patients whose progressive disease was correctly predicted and the time of that prediction. The evaluation was performed (1) by using actual biomarker data and (2) by using data that simulated missing data points from the Bayesian model. The performance of different biomarkers was compared to explore their biological roles in tumor progression.

**Characterizing biomarker trajectories during treatment**. In this study, we applied a Bayesian hierarchical modeling approach to investigate concentrations of a specific biomarker as a function of treatment time. The approach was applied to develop probability models that define joint probability distribution for all observed and unobserved data[49]. This approach was especially suitable for analyzing biomarker data as we present in this study because of its flexibility to produce complicated models with multiple conceptual layers and large numbers of parameters, as well as its natural framework to tolerate missing data points which occur frequently in clinical trials. A similar modeling approach was applied in our previous study investigating biomarker data from ovarian cancer patients[12], in recent literature for modeling trajectories of PSA in prostate cancer[50] and multivariate clinical factors in Parkinson's disease[51].

The Bayesian hierarchical model, once developed, enabled us to address two questions. Firstly, it provided a quantitative framework to define the dynamics of biomarker concentrations over time. It allowed justification of our hypothesis on inflection point, i.e., an inflection point separates the decreasing part of biomarker trajectory from the subsequent increasing part and the inflection point can be considered as a sign of changing tumor behavior. Secondly, the model enabled inference of the concentration of a biomarker, on any patient and at any time point during treatment. This meant that we could carry out pseudo-trials on the same cohort of patients but on different sample collection schemes. This inference process was different from the concept of prediction, as it did not generate data for a new patient based on an existing patient. Instead such estimation resembles interpolating values for missing data points utilizing information from individual patients and the whole population. Pseudo-trial data can be used to determine optimum rules for biomarkers to predict tumor progression, which will be described in detail below.

**Bayesian model and Markov Chain Monte Carlo modeling**. The Bayesian hierarchical model was set up based on a piecewise linear relationship between biomarker quantity (concentration for circulating biomarkers; various units for imaging biomarkers) and treatment time ($t$), parameterized as follows:

$$C_{\text{biomarker}}(t) = \alpha + S(t_{\text{inflection}} - t)\beta t$$
$$+ S(t - t_{\text{inflection}})(\beta t_{\text{inflection}} + \gamma(t - t_{\text{inflection}})) + \varepsilon \quad (1)$$

where $C$ represents the quantity of the biomarker being modeled, $\alpha$ is the pre-treatment concentration of the biomarker, $t_{\text{inflection}}$ is the inflection point of biomarker trajectory, $\beta$ is the slope before the inflection point, $\gamma$ is the slope after the inflection and $\varepsilon$ is a random error. $S$ is an indicator function where

$$S(x) = \begin{cases} 1 & \text{if } x > 0 \\ 0 & \text{if } x \le 0 \end{cases} \quad (2)$$

The parameters were assigned the following prior distributions to follow the structure of a Bayesian hierarchical model:

$$\log(\alpha) \sim N(\mu_\alpha, \sigma_\alpha^2)$$
$$\beta \sim \alpha \times (\beta_1 + E_1 \times \beta_2)$$
$$\beta_1 \sim N(\mu_{\beta 1}, \sigma_{\beta 1}^2)$$
$$\log(\beta_2) \sim N(\mu_{\beta 2}, \sigma_{\beta 2}^2)$$
$$E_1 \sim B(1, p_1)$$
$$\gamma \sim \alpha \times (\gamma_1 + E_2 \times \gamma_2)$$
$$\gamma_1 \sim N(\mu_{\gamma 1}, \sigma_{\gamma 1}^2)$$
$$\log(\gamma_2) \sim N(\mu_{\gamma 2}, \sigma_{\gamma 2}^2)$$
$$E_2 \sim B(1, p_2)$$
$$\varepsilon \sim C \times N(0, \sigma_\varepsilon^2)$$
$$t_{\text{inflection}} \sim U(21, \text{PFS} - 21)$$

Specifically, $\alpha$ follows a log normal distribution, $\beta$ is dependent on $\alpha$ and both $\beta$, $\gamma$ are modeled to follow a combination of a normal and a log normal distribution that controlled by $E_1$ and $E_2$ (Bernoulli distributions). The reason for setting $\beta$ and $\gamma$ as above is to accommodate the fact that Tie2 trajectories can be clustered into two distinct groups (Fig. 3) that have different $\beta$ and $\gamma$ values ($\beta_1$, $\beta_2$ and $\gamma_1$, $\gamma_2$). $\varepsilon$ can be considered as technical variation and is therefore modeled to be dependent on biomarker quantity. $t_{\text{inflection}}$ is uniformly distributed between 21 days after treatment starts and 21 days before diagnosis of progression.

The parameters were assigned appropriate prior distributions in accordance with the biomarker being modeled. For example, the following prior values were assigned when Tie2 was modeled:

$$\mu_\alpha \sim N(4.3, 1)$$
$$\sigma_\alpha \sim N(0.6, 1)$$
$$\mu_\beta \sim N(-2, 1)$$
$$\mu_{\beta 2} \sim N(1.5, 1)$$
$$\sigma_\beta \sim N(1, 1)$$
$$\sigma_{\beta 2} \sim N(1, 1)$$
$$p_1 \sim U(0.2, 0.7)$$
$$\mu_\gamma \sim N(0, 1)$$
$$\mu_{\gamma 2} \sim N(1, 1)$$
$$\sigma_\gamma \sim N(2, 1)$$
$$\sigma_{\gamma 2} \sim N(1, 1)$$
$$p_2 \sim U(0.2, 0.7)$$
$$\sigma_\varepsilon \sim N(0.15, 10)$$

All biomarkers being modeled were transformed to control their dynamic ranges. For example, M65 was squared and then multiplied by 10. Ang2 was log$_2$ transformed, and its product with Tie2 was further divided by 1000.

The posterior distributions of the parameters were determined using an Markov Chain Monte Carlo (MCMC) approach as implemented in Winbugs 1.4. For each model, three Markov chains were trained at a same time to ensure best coverage.

They were updated for 50,000 iterations or until sufficient evidence of model convergence was observed, whichever occurred later. A converged model will be updated for a further 100,000 iterations to estimate posterior distribution of each model parameter. According to our observations, convergence was achieved typically within 20,000 update iterations.

**Bayesian hierarchical model generates pseudo-trial data.** The Bayesian hierarchical models parameterized biomarker trajectories during treatment for each individual patient by means of posterior probability distribution, considering the observed biomarker data as a random sample from these distributions. Naturally it allowed "pseudo-trial" data to be generated via Bayesian predictive inference, which represented the model's best guess of biomarker concentration for a specific patient at a given time, as if the trial were repeated on the same patients and data were collected based on new collection scheme. It should be noted that the prediction process did not create new pseudo-patients, instead it resembles interpolation of values of missing data points.

**Predicting tumor progression based on modeled inferred data.** We intended to develop rules on how to use the selected biomarkers in clinic, that is, how to provide early predictions of tumor progression by monitoring the biomarkers sequentially during treatment. Based on the inflection point hypothesis that was described above, we designed a generic rule that prediction on tumor progression was considered if elevation of a biomarker with respect to its recorded nadir point exceeds a designated alarm threshold. The optimal threshold can be determined using the following steps:

(1) For any threshold we generated pseudo-trial data for a given biomarker via prediction from the Bayesian model at a designated time. In this study a monthly sample collection interval ($30 \pm 5$ days) was applied.
(2) For each patient, we examined the recorded data sequentially following treatment time. For any given time point, we calculated the percentage change of biomarker concentration against the nadir concentration recorded prior to this time point. If the percentage change was larger than the designated threshold value a prediction of tumor progression was considered to take place at that time point.
(3) We repeated steps 1–2 five thousand times.
(4) Biomarker prediction performance was evaluated by summarizing all the recorded prediction times.

The threshold that predicts disease progression for as many patients as possible, at a time reasonably close to the genuine date of progression was considered as the optimum threshold (Table 4).

## Data availability
Data generated in this study are available in the Mendeley repository https://doi.org/10.17632/xcsbspcghg.2.

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

## Acknowledgements
We would like to acknowledge the support of the ECMC (grant reference C480/A15578), the support of Cancer Research UK to the Manchester Cancer Research Centre (C147/A18083) and to the CRUK Manchester Institute (C5759/A20971), and the NIHR Clinical Research Facility, the Manchester NIHR Biomedical Research Centre and CRUK/EPSRC Imaging Centre Award (C8742/A18097). Cancer Research UK funds J.P.B.O'.C. (C19221/A22746). The translational research in the manuscript was funded through an unrestricted grant from Roche. The clinical trial, biomarker schedule and analysis were planned, conducted, and analyzed by the authors. The Manchester Academic Health Sciences Centre (MAHSC) Clinical Trials Unit managed the trial, with 3-monthly reviews by the safety monitoring committee. The trial was a single centre study, where the costs for the laboratory staff, clinical fellows, excess treatment costs, as well as bevacizumab were provided by Roche (Basel, Switzerland) to the University of Manchester, UK. Specialized circulating biomarker laboratory infrastructure, senior scientific oversight of the laboratory analyses of circulating biomarkers and quality assurance of the resultant data were funded via CRUK core funding to the Cancer Research UK Manchester Institute (CD). Specialized imaging biomarker laboratory infrastructure, senior scientific oversight of the imaging biomarker analyses, and quality assurance of the resultant data were funded via CRUK cancer imaging centre funding to the University of Manchester (J.P.B.O'.C., G.J.M.P.). Roche was informed of the study progress and results and has received a copy of the final manuscript. Roche has reviewed the final manuscript but the authors determined the final content. The corresponding author (G.C.J.) has overseen the analysis of all data and is responsible for the conduct of the study. All authors have approved the final manuscript.

## Author contributions
G.C.J. oversaw the entire project from concept, protocol writing, conduct of the study, analysis, and manuscript writing. C.Z. and G.C.J. analyzed all of the data presented in the manuscript and oversaw writing of the manuscript. A.B. conducted the ELISA analyses throughout the project and assisted with manuscript preparation. L.H. K.M-.M., D.S. and N.M. conducted the trial and oversaw imaging studies and sample collection. A.C., M.P.S, J.W.V., S.M., M.B. and J.H. oversaw recruitment and patient management and were involved in protocol revisions. D.Mc.E. led the research nurses and was responsible for coordinating sample collection and imaging studies. K.S. led the blood biomarker laboratory team. R.A.L., Y.W., S.C. and C.R. were responsible for application of DCE-MRI protocols and definition of regions of interest. P.M. analyzed the anatomical CT scans and identified areas for analysis on anatomical CT scans that were then evaluated by DCE-MRI. S.J.S. and O.d.P. led the industry collaboration supporting the project from concept, clinical protocol development, and delivery through to analysis and presentation. A.J., J.P.B.O'.C., and G.J.M.P. oversaw the advanced imaging, from acquisition, standardization, data acquisition, analysis, and Q.C. standards. C.D. is director of the circulating biomarkers laboratory and oversaw the conduct and analysis of all ELISAs. The manuscript was written by G.C.J., C.Z., J.P.B.O'.C., G.J.M.P. and C.D. and all authors have approved it.

## Additional information

**Competing interests:** G.J.M.P. is a director of and shareholder of Bioxydyn, a company spun-out from the University of Manchester with an interest in imaging biomarkers. G. C.J. and C.D. have received grant funding for this project from Roche, which was administered through the University of Manchester. G.C.J. has attended advisory boards for Roche. J.P.B.O'.C. and C.D. declare no competing interests relevant to this manuscript. All remaining authors declare no competing interests.

