## [Peer Review File · Nature Communications]

Reviewers' Comments:

Reviewer #1:

Remarks to the Author:

In this manuscript Jayson and colleagues address a crucial issue in that they search for reliable biomarkers of Bevacuzimab response. The comparison of a range of biomarkers strengthened the case for Tie-2 as a circulating biomarker for response. The authors have done a good job in analysing the distinct biomarkers and have adequately modelled the different parameters.

In addition, I feel that the authors have done a good job in answering the questions of the previous reviewers.

The only minor revision I would suggest is to explain the choice of analysis a bit better. The modelling and underlying mathematics are relatively complex and it would help the uninformed reader to get a better idea on the reason behind the current analyses.

Reviewer #2:

Remarks to the Author:

Comments on the revised manuscript

The revised manuscript very nicely describes in more details the approach taken and provides high level summaries of the data. When reading the paper I am left with more questions and a slight frustration that I can't properly get to grips with what is a very valuable data set. Despite improvement it still does not really provide the right level of clarity and depth to the data to allow the reader to dig into the methodology and findings. Given the degree of complex modelling that delivered the associations I would have expected the authors to take time to introduce more detailed explanation of their approach and walk the reader through the rationale in more detail. Providing more detail may be challenging but would help de-mystify some elements. Modelling is fantastically useful as a tool, but to be deployed has to be carefully articulated.

In answering the reviewers' comment the rebuttal letter seeks to address most of the comments, but in many areas there does not seem to be incorporation of the comments into the discussion, which seem extremely brief given the importance of the work.

The pivotal piece of the data is the robustness of the change in plasma Tie-2 levels. The guidance that authors give for the application of this new biomarker appears to equate to a change from 2ng/ml to 1.5ng/ml (25%) and then a rise of 30 percent back to c.2ng/ml as the patient progresses. To build confidence in the ability to do this in other trials then it would be important to provide the reader with more information on the performance of the biomarkers. For example how do the errors on the assay perform (assuming it is performed in duplicate or triplicate per sample); what is the variation between the two baseline samples across individual patients. Could examples of the critical biomarkers in specific patients be supplied to illustrate what is happening. It is often more convincing to see these effects at a population rather than patient level. This is not to question the validity of the finding but rather to provide a means to enable adoption more easily into others' work.

The same could be provided for CK18.

The discussion needs more bringing into it as outlined in many comments.

Specific comments

Page 4 lines 113 -119 and Table 3 describe prognostic biomarkers associated with outcome. Are high or low representations of these biomarkers associated with poor outcome?

Page 6 line 155 – while it is clear that pTie2 at baseline and first treatment is related to vascular changes, it is not that easy to make the same conclusion on progression as it is possible that changes on progression could also be associated with increases in Tie-2 positive macrophage-like cells (as shown by the De Palma and other labs). The authors allude to this in the results but should address it more in the discussion as it is a limitation of the conclusions.

Of the patients that fail to respond in the first cycles by reducing Tie-2 have the authors considered that they may also exhibit high baseline Tie-2 positive macrophages?

On reflection it is odd that Ang-2 appears to show the same pattern of modulation as plasmaTie-2. Can the authors comment on whether this may help as part of a composite biomarkers?

Table 2 As presented Table 2 does not really help understand what is happening. Do all these markers do down or up? There are no units, pg or ng? Tie2 is on the Table but just as Tie2?! is it possible to indicate the changes that occur in the samples in the VEGFi phase alone (as represented in Figure 2?) and in the VEGFi Chemo phase. Minor point – it is a shame that VEGFR3 is missing it is another potential vessel marker that changes on VEGFi treatment and would be interesting to see how it correlates with plasmaTie2 on response and then progression.

Page 7 line 182 refers to non-VEGF dependent pathways it would be worth taking a little time to expand on this explanation a little in the discussion to support why this conclusion is drawn in more detail.

Figure 2a and 2b Refers to the biomarker relationships at baseline and on treatment. From the data presented is it clear that other modellers could reproduce or pressure test the modelling with this representation and the description in the literature

Will the totality of the data set be deposited / made available?

Minor point

The acronym pTie2 is still misleading in the view of this reviewer. Other plasma born receptors are referred to as the soluble version of the receptor. It would be extremely helpful to have a better term.

Supplementary data

Supplementary Table 1 It is assumed that the numbers in the table refer to numbers of patients. Could the columns be labelled more clearly.

Supplementary Table 2 this is helpful however were the circulating biomarkers on day 3 analysed or included in this analysis, most of the graphs do not have a day 3 data point. This same point applies to all the graphs in the supplementary figures.

Supplementary Fig S1a could the units be referred. It would be very interesting to represent the VEGF I alone phase on the plots e.g. with a dotted line

Supplementary Fig S1b switches to ratios again. However it would be important to also comment on some of the other changes in this cohort of biomarkers, e.g. is VCAM going up for a reason? IL8 and SDF, IL6 are commonly induced on combination therapy – interesting that they are not changing in the VEGFi alone phase?

Supplementary Fig S1C would be worth a little more discussion of the other biomarker data in these figures. Is it reasonable to assume that the variation in the biomarkers is important. Other biomarkers clearly associated with progression, again at least drawing some attention to these other biomarkers would be of value. Out of interesting do any of the biomarkers listed in Fig S1c change independent of progression.

Discussion

There is no assessment of the limitations of the study, or an attempt to address areas where the authors conclusions could have a different explanation.

The discussion does not really discuss how this could approach could be implemented in a trial.

The correlation between pTie2 and the imaging data does not demonstrate that the plasma Tie-2 is reflective of reduction in vessels it implies or strongly suggests. A formal pre and post biopsy study would be required to demonstrate this point. Moreover it is possible that the pTie-2 is not all tumour derived, however the imaging assessment focuses only on the tumor. The discussion as present asserts and simple association that in reality is likely to be more complex?

Line 260 – 262 refers to the macrophage issue, it would be helpful to expand this part of the discussion.

Comments on the revised manuscript

Reviewer #1 (Remarks to the Author):

In this manuscript Jayson and colleagues address a crucial issue in that they search for reliable biomarkers of Bevacuzimab response. The comparison of a range of biomarkers strengthened the case for Tie-2 as a circulating biomarker for response. The authors have done a good job in analysing the distinct biomarkers and have adequately modelled the different parameters.

In addition, I feel that the authors have done a good job in answering the questions of the previous reviewers.

The only minor revision I would suggest is to explain the choice of analysis a bit better. The modelling and underlying mathematics are relatively complex and it would help the uninformed reader to get a better idea on the reason behind the current analyses.

Throughout the revised manuscript we have attempted to explain the analysis and the rationale for the different analytical techniques. This has already been included in the revised, submitted version in response to a similar question from another reviewer.

Reviewer #2 (Remarks to the Author)

We would like to thank the reviewer for his/her very helpful comments, which have led to expansion of many points throughout the manuscript. In particular we feel that the reviewer's request to include individual patient data was very valuable and that this has added to the manuscript. All amendments that have been requested by the referee are shown in red type face.

The revised manuscript very nicely describes in more details the approach taken and provides high level summaries of the data. When reading the paper I am left with more questions and a slight frustration that I cant properly get to grips with what is a very valuable data set. Despite improvement it still does not really provide the right level of clarity and depth to the data to allow the reader to dig into the methodology and findings. Given the degree of complex modelling that delivered the associations I would have expect the authors to take time to introduce more detailed explanation of their approach and walk the reader through the rational in more detail. Providing more detail may be challenging but would help de-mystify some elements. Modelling is fantastically useful as a tool, but to be deployed has to be carefully articulated.

Throughout the text we have tried to explain the rationale for the different analytical steps. Further, we propose to publish the entire data set on line with the paper, if the journal can accommodate this. Alternatively, we will make the entire data set available through the University of Manchester.

In answering the reviewers comment the rebuttal letter seeks to address most of the comments, but in many areas there does not seem to be incorporation of the comments into the discussion, which seem extremely brief given the importance of the work.

We have significantly expanded the discussion to cover the relevant areas.

The pivotal piece of the data is the robustness of the change in plasmaTie-2 levels. The guidance that authors give for the application of this new biomarker appears to equate to a change from 2ng/ml to 1.5ng/ml (25%) and then a rise of 30 percent back to c.2ng/ml as the patient progresses. To build confidence in the ability to this in other trials then it would be important to provide the reader with more information on the performance of the biomarkers. For example how do the errors on the assay perform (assuming it is performed in duplicate or triplicate per sample); what is the variation between the two baseline samples across individual patients. Could examples of the critical biomarkers in specific patients be supplied to illustrate what is happening. It is often more convincing to see these effects at a population rather than patient level. This is not to question the validity of the finding but rather to provide a means to enable adoption more easily into others work. The same could be provided for CK18.

We have supplied references on the validation work that was used to establish the assay; we have added figures (fig S2 and S3) on technical variation and have provided single patient data from several patients (Fig 6b-e), which show the distinction between epithelial and vascular, response and progression. We have also referenced (reference 36) the work that we performed to demonstrate the reproducibility, stability and validity of the multi-plex ELISA used for the soluble biomarkers. Both the blood biomarker and imaging biomarker laboratories are run to the standards of GCP-L.

The discussion needs more bringing into it as outlined in many comments.

The discussion has been significantly expanded.

Specific comments

Page 4 lines 113 -119 and Table 3 describe prognostic biomarkers associated with outcome. Are high or low representations of these biomarkers associated with poor outcome?

In table 3, high biomarker levels were associated with worse outcome. The specified section described a uni-variate and a multi-variate proportional hazard regression analysis with the results summarised in Table 3. The uni-variate analysis was designed to select a subset of biomarkers for subsequent multi-variate analysis. Therefore we only reported p-values for the selected subset of biomarkers. For the multivariate analysis, p values, hazard ratios (HR) and their 95% CI were reported, consistent with the REMARK guidelines. As summarised in Table 3 the HR of all prognostic biomarkers were larger than 1 indicating that a high level of biomarkers was associated with worse outcome.

We have updated the Methods section to clarify the logic of the proportional hazard regression analysis, "Biomarkers with uni-variate p-values smaller than 0.05 were selected for subsequent multi-variate analysis...". We have also updated the Results section to facilitate model interpretation "High levels of these biomarkers associated with reduced PFS". The table legend has also been updated to explain these findings.

Page 6 line 155 – while it is clear that pTie2 at baseline and first treatment is related to vascular changes, it is not that easy to make the same conclusion on progression as it is possible that changes on progression could also be associated with increases in Tie-2 positive macrophage-like cells (as shown by the De Palma and other labs). The authors elude to this in the results but should address it more in the discussion as it is a limitation of the conclusions.

We have included a review of this issue particularly in the discussion. We agree that published data point towards macrophages mediating the progression effect and have written a paragraph in the discussion highlighting this possibility. Nevertheless, we feel that such a putative mechanism does not limit the conclusions around Tie2 but would warrant further studies at the tissue level.

Of the patients that fail to respond in the first cycles by reducing Tie-2 have the authors considered that they may also exhibit high baseline Tie-2 positive macrophages ?

We agree that this is an excellent hypothesis that should be investigated. While we agree that published literature point to acquired resistance to VEGFi being mediated by macrophages, it is not clear that innate resistance is effected through the same mechanism.

On reflection it is odd that Ang-2 appears to show the same pattern of modulation as plasmaTie-2. Can the authors comment on whether this may help as part of a composite biomarkers?

The reviewer is right that Ang2 and Tie2 share similar longitudinal patterns during treatment (strong correlation). To investigate this, we created a conceptual biomarker, Ang2 times Tie2, and investigated if the conceptual biomarker provided improved sensitivity/specificity with respect to predicting tumour progression. A marginal improvement was observed in simulation-based analysis but not in the actual data. We therefore concluded that including Ang2 would not improve the clinical utility of Tie2, while generating a biomarker that was much harder to use. Although unproven, we surmise that the reason for the parallel modulation of Ang2 and Tie2 is that the two molecules probably exist in the circulation as bound ligand-receptor complexes. We haven't added this particular point into the discussion because, although interesting, there are many hypotheses that can be generated about the data of which this is one example. We will, however, publish the entire data set either through the journal (if accepted and feasible) or through the University of Manchester.

Table 2 As presented Table 2 does not really help understand what is happening. Do all these markers do down or up? There are no units, pg or ng? Tie2 is on the Table but just as Tie2 ?! is it possible to indicate the changes that occur in the samples in the VEGFi phase alone (as represented in Figure 2?) and in the VEGFi Chemo phase. Minor point – it is a shame that VEGFR3 is missing it is another potential vessel marker that changes on VEGFi treatment and would be interesting to see how it correlates with plasmaTie2 on response and then progression.

We have rewritten the legend to the table and hope that this is now more useful. As stated in the legend, the changes of biomarkers were listed as percentage of pre-treatment levels. Therefore values above 100 indicated elevated biomarker levels while values below 100 indicated reduced biomarker levels. We have revised the table legend to clarify this issue. It now states "Pre-treatment levels of circulating and imaging biomarkers were summarized by their median and standard deviation. Median changes of each circulating and imaging biomarkers at three time points during treatment are summarized with reference to the percentage of pre-treatment biomarker levels. Therefore values above 100 indicate elevated biomarker levels while values below 100 indicated reduced biomarker levels. Plots of biomarker changes are shown in Figure S1."

The units for soluble biomarkers are pg/ml. Table 2 and Fig S1 includes these. The changes of biomarkers at the end of the single agent VEGFi phase (day 14) were summarised in the 4th column of the table, while changes during the whole treatment course were summarised in the 5th column.

Finally, we agree that it would have been nice to look at VEGFR3 but when the study was established we had to choose the most promising candidates for evaluation.

Page 7 line 182 refers to non-VEGF dependent pathways it would be worth taking a little time to expand on this explanation a little in the discussion to support why this conclusion is drawn in more detail.

We have expanded this section to try to explain the hypothesis.

Figure2a and 2b Refers to the biomarker relationships at baseline and on treatment. From the data presented is it clear that other modellers could reproduce or pressure test the modelling with this representation and the description in the literature Will the totality of the data set be deposited / made available?

We will submit the entire data set for publication either through the journal, if accepted for publication, or through the University of Manchester

Minor point

The acronym pTie2 is still misleading in the view of this reviewer. Other plasma born receptors are referred to as the soluble version of the receptor. It would be extremely helpful to have a better term.

We have used Tie2 as the simplest abbreviation for plasma Tie2 concentrations throughout the manuscript.

Supplementary data

Supplementary Table 1 It is assumed that the numbers in the table refer to numbers of patients. Could the columns be labelled more clearly.

The table legend has been revised to “The table lists the number of patients within categories defined by pre-treatment [VEGF-R2/Ktrans] and Tie2 trajectories during treatment.....”. We have also quoted Figure 3 in the table legend to clarify how the categories described in the table were created.

Supplementary Table 2 this is helpful however were the circulating biomarkers on day 3 analysed or included in this analysis, most of the graphs do not have a day 3 data point. This same point applies to all the graphs in the supplementary figures.

We have double checked our data and found although plasma samples were collected for day 3 they were not analysed. The table has been corrected accordingly.

Supplementary Fig S1a could the units be referred. It would be very interesting to represent the VEGF I alone phase on the plots e.g. with a dotted line

While the dotted line idea is excellent we were concerned that the graphs were already small and therefore that the dotted lines would obscure the data. We therefore stated the single agent bevacizumab administration period in the figure legend.

Supplementary Fig S1b switches to ratios again. However it would be important to also comment on some of the other changes in this cohort of biomarkers, e.g. is VCAM going up for a reason? IL8 and SDF, IL6 are commonly induced on combination therapy – interesting that they are not changing in the VEGFi alone phase?

We saw significant patient-to-patient variations in many biomarker concentrations, as shown in Table 2. Log ratios were applied to reduce such variations. We felt that although the points raised by the reviewer were of great interest, the complexity of the data and the significant inter-patient variation would have led to a large discussion while the manuscript really warranted an element of focus to avoid distracting the reader.

Supplementary Fig S1C would be worth a little more discussion of the other biomarker data in these figures. Is it reasonable to assume that the variation in the biomarkers is important. Other biomarkers clearly associated with progression, again at least drawing some attention to these other biomarkers would be of value. Out of interesting do any of the biomarkers listed in Fig S1c change independent of progression.

We agree with the reviewer that the dataset has potential beyond the scope of this paper. The purpose of the study was to seek biomarkers that related to vascular response and vascular progression following VEGFi. Of particular interest is the performance of the Ang-Tie system as we have discovered the potential of Tie2 as a vascular progression biomarker in ovarian cancer. In this study we independently identified the Tie2 signal to be of vascular origin and therefore strategically decided to focus in-depth on Tie2. The discussion of other biomarkers was limited to avoid diverting

the reader away from the key message concerning Tie2. We hope that by publishing the entire data set other investigators can analyse the data according to their own hypotheses and interests.

Throughout the study we tried to take an unbiased approach to avoid the confounding feature of multiple comparisons. Thus the network and cluster analyses were unbiased and focused our attention on Ang/Tie rather than the other biomarkers. Further, as we showed with the Tie2 cluster analyses, the concentration-time curves shown in Figure S1c represent the average change for the entire group and therefore subgroup analysis would be relevant. If all patients manifest the same change in a biomarker then the clinical value of a biomarker is low, even if biologically it is of interest. We will add arguments along these lines to the discussion.

Discussion

There is no assessment of the limitations of the study, or an attempt to address areas where the authors conclusions could have a different explanation. The discussion does not really discuss how this could approach could be implemented in a trial.

We have derived vascular response and vascular progression criteria, which have been included in the paper and in particular have added a section on precision medicine at the end of the results section to show how individual data can be used to manage patients. We have also added text discussing the limitations of the study in the discussion.

The correlation between pTie2 and the imaging data does not demonstrate that the plasma Tie-2 is reflective of reduction in vessels it implies or strongly suggests. A formal pre and post biopsy study would be required to demonstrate this point. Moreover it is possible that the pTie-2 is not all tumour derived, however the imaging assessment focuses only on the tumor. The discussion as present asserts and simple association that in reality is likely to be more complex? Line 260 – 262 refers to the macrophage issue, it would be helpful to expand this part of the discussion.

We have added text on macrophages and the need for a tissue based study to define the cellular mechanisms.

Response to referee #2's comments on the previous point-by-point response

Colour code for responses

Black text = original comment from reviewer

Blue text = our original reply to the reviewers

Red text = reviewer's consideration of our responses

Green text = current response to remaining questions

Referees' comments: Referee

#2

The manuscript by Jayson et al seeks to give insight into the clinical use anti-angiogenic drugs targeting VEGF, and ways to gain insight into the ongoing clinical benefit a patient is receiving. This is an important topic as VEGF inhibitors of different classes are widely used in the clinic. Therefore, biomarker strategies that inform on whether a patient is gaining benefit, and when that benefit is lost could be extremely valuable. Many studies have sought to address this issue but to the knowledge of this reviewer this is the first bespoke study that has sought to identify accessible biomarkers linked to progression following VEGF inhibitor treatment assessing the tumour and non-tumour response. The study describes a longitudinal analysis of a 70-patient clinical study where mCRC patients received bevacizumab in combination with chemotherapy. Using a range of biomarker analyses the study establishes a theoretical model that simultaneously describes the contribution of events in the epithelial and non-epithelial compartments of the tumour to response and subsequent progression on therapy.

This is certainly a very interesting concept highlighting the importance of the Tie2 signalling axis in response following treatment. The study builds on elements described in earlier papers in ovarian cancer from the same lab demonstrating a link between soluble Tie2, Ang2/1 and clinical response or progression.

We are grateful for the reviewer's positive comments

Major comments

The analysis is thought provoking, however by virtue of compressing the data into a letter format much of the information that would be of real interest is lost.

We were asked to comply with the word restrictions of *Nature Letters*, which radically restricted the space available to write up the manuscript. Consequently, we have completely re-written the manuscript.

Moreover, the lack of detail makes it very challenging to consider how the data can inform on other studies investigating the use of VEGF signalling inhibitors. The obvious critique is that this is performed on one clinical study, drawing parallels with other supporting data would be important (e.g. previous work in ovarian cancer from the same group)?

We have tried to emphasise in our introduction and then discussion that (i) our ovarian study discovered and validated plasma Tie2 as a response biomarker for bevacizumab in ovarian cancer and (ii) that the current paper validates the concept in the best possible way i.e. through validation of the core finding in a different disease, colorectal cancer, while also extending the understanding of the biomarker signal through advanced imaging and its analysis; (iii) we then highlight that review of other literature shows that in colorectal cancer, glioma and gall bladder cancer patients treated with the VEGF RTKi, cediranib, the same pharmacodynamic changes in pTie2 have been reported although their clinical significance was not analysed in the way presented here. Thus, we can be confident for the first time that we have a validated, tumour vascular biomarker for VEGF inhibitors that has been confirmed in two different classes of drugs that inhibit VEGF and in at least two different laboratories.

Reviewer comment: Agree with this approach, the statement could also be softened to recognise that this is a pharmacodynamic marker VEGFi as a class, as discussed above others

will have to prove the modulation of plasma Tie-2 originates from the tumour vasculature alone with biopsy studies. Plasma-Tie-2 may not only be derived from endothelium in the progression phase?

In the latest revision, we have discussed the concept of pharmacodynamic modulation of Tie2 and other molecules in the introduction. Then in response to this reviewer and the other reviewer we have introduced an additional section in the discussion (bottom of page 14 extending into page 15) where we discuss the potential role of macrophages as the source of Tie2 and, as stated by the reviewer here, that a tissue-based study would be required to resolve the mechanism.

There are some very strong conclusions being drawn from this exploratory analysis therefore it would seem important to show the patient data in more detail to enable the reader to form their own views. My major concern is that as the data is presented it is not accessible for the reader, and is perhaps over calculated, the focus of the paper is on the modelling approach which seems to detract from the overall value of the data set.

The manuscript has been completely re-written with the information returned into the manuscript, which was considerably shortened to fit the *Nature Letter* format. The reviewer is correct that we do present the modelling but this is because modelling of such data is very novel at the moment and in the UK, the Medical Research Council is currently funding research that evaluates data in this way. The reviewer is correct to highlight that the modelling is being presented but that is exactly because it is novel. He/She is not correct about the data being over-calculated since, as stated below, we use an unsupervised analytical approach to avoid such bias.

Reviewer comment: The authors mis-read the intention of the comment. The concern is that as presented it is hard for others to dig into the modelling in great detail due to the granularity of the data presented. While modelling approaches are extremely valuable the outputs are often taken as absolute, and the caveats are not recognised. The approach is great, from the granularity presented could others reproduce this work? The value in the data set is the breadth of the monotherapy and combination therapy biomarkers collection and the conclusions you draw from them. Others looking at the same data may generate different hypotheses or conclusions.

We are grateful for the reviewers' comments and felt that the only way that we could be entirely open about the data was to publish the entire data set. The University of Manchester has agreed to this and so, if the manuscript is accepted, we will make the data set publicly available.

The soluble biomarker data is all normalised to a baseline and then expressed as a percentage change. It would be much more informative to have the raw data presented within the context of the study. The manuscript stimulates a range of questions that are hard to answer as presented

We have added in raw data and added a new table (table 2), which summarises the raw biomarker data and their behaviour during the trial.

Reviewer comment: data presented is more informative, it would be good to get better reference to units on the figures etc. As discussed above there are two pre-treatment samples from which the variation and 95% confidence intervals are calculated. It would be informative to see the patterns of the patients' biomarker changes over these two baseline samples (even a subset of patient) and the first two monotherapy bev plasma samples.

As mentioned above we will publish the entire data set, if the manuscript is accepted for publication. This data set will include all of the requested information. However, I hope that most of the information requested by the reviewer is now included in the revised table 2 (page 18) where in the 3rd column from the left we show the median value of the biomarker with its units +/- the standard deviation.

To what degree are these biomarkers changing? The values imply that the changes are roughly a half of the initial concentration? What is the baseline concentration what is the nadir and steady state suppression?

There is a range of variation in biomarker parameters according to each biomarker. To address this, a new table has been added to the manuscript (table 2) that summarises the biomarker values before treatment, the acute changes at day 14 following single agent bevacizumab administration, the largest changes recorded during treatment and those seen at progression. We have also generated a new figure (sFigure 1) that includes raw biomarker data to illustrate changes of these biomarkers during treatment.

Reviewer comment: Table 2 and supplementary figure 1 help a lot as stated above

Is there a large or small range in the biomarker concentrations at baseline and on response? All of our biomarker studies incorporate reproducibility studies before intervention to establish the coefficient of variation (CV) of each biomarker. These data are then used to define the 95% confidence intervals that allow us to state whether a therapeutic agent has impacted on biomarker behaviour or not. In general terms, biomarker concentrations did vary over time, post-therapy, typically exceeding 100% percentage CV and reaching a

maximum of 5000%. Overall we found that better consistency was achieved through looking at biomarker changes rather than concentrations. Accordingly we chose to normalise the concentration of these biomarkers against their pre-treatment levels.

Reviewer comment: this makes sense, it would however help to spell this out carefully in the main body of the paper particularly around the performance of the critical biomarkers.

I hope that the revisions to Table 2 (page 18) provide this information. The variance of the biomarkers pre-treatment is shown in the 3rd column from the left.

How do early progressor and late progressor biomarker profiles compare?

A key question that we tried to address in this study is how differences in biomarker profiles associate with tumour progression. In this study, patients were categorised using an unsupervised clustering approach according to their biomarker profiles. The PFS between the two patient groups was then compared (figure 3). By using this approach, we bypassed the challenge of over-fitting.

Had we followed the methods implied by the reviewer's question this would have led to over-fitting through arbitrary definitions of early and late progression. Such an approach would have incurred the confounding factors of performance status and tumour size that would have been associated with different progression times but not with vascular progression.

Reviewer comment: While it is agreed that the approach taken removes the bias leading the identification of plasma Tie2 the next natural question having established the Tie-2 relationship is whether all the other biomarkers studied tracked pTie2 or were independent of pTie2 in these subsets. The question is not intended to suggest a different analytical method. The reason for the curiosity is that a number of the markers measured have been associated with progression / resistance in other studies and is there learning or cross over to explain other findings. The comment is stimulated by the interesting observation that certain other biomarkers such as IL6 SDF, VCAM change the relationship following treatment in the context of the model presented in the figure 3. Could the authors comment on these markers?

There are two important points here: First, we focused on Tie2, Ang2 and K^{trans} because these were the only biomarkers that became correlated through the effect of single agent bevacizumab. Nearly all other biomarker relationships were disrupted by treatment with bevacizumab. Second, although not shown here, V-CAM behaved in a very homogeneous fashion and as a consequence there was very little variation between patients that would allow identification of subgroups of patients who benefitted or who didn't benefit from bevacizumab. As many such hypotheses could be generated, we felt it best not to go into this issue in detail in the manuscript but instead to

publish the data so that others can interrogate the data set to address their own hypotheses. I can confirm that the University of Manchester has agreed to publish the data set, if the manuscript is accepted for publication.

Do all patients receive chemo throughout the study or do some of the progressions occur in patients on bev alone?

I am not exactly clear to which component of the trial protocol the reviewer is referring. All patients were initially treated with single agent bevacizumab for 2 weeks to elicit a biomarker signature. No patient developed progressive disease during that period. They then received cytotoxic chemotherapy with bevacizumab until progressive disease occurred. As set out in the trial protocol and described in the Methods section patients were able to continue combination cytotoxic chemotherapy/bevacizumab until progressive disease developed. The earliest point at which a patient developed progressive disease was 46 days although to some extent this is related to the internationally accepted minimum period of 6 weeks treatment before carrying out the first response-assessment scans.

Reviewer comment: This has answered the question. No patients had progressed by the time of the third scan day 15.

While it is appreciated that it is hard to present details in the current format it would be very important to see the data in a form that enables the reader to understand what is happening in the study.

We have completely re-written the paper and have added a table 2 and sFigure 1 to show raw biomarker data.

Reviewer comment: helpful but please see other comments

We will publish the entire dataset through the University of Manchester, if the manuscript is accepted for publication.

The demo table appears to indicate a range of severity/condition status among the 70 patients, how relevant is the % of patients with different baseline status for the profile of change in biomarker post-treatment, and/or for PFS, let alone for the relationship between biomarker

change and progression?

There was no significant association between pTie2 profiles and established clinical demographic factors. The only variable that correlated with plasma Tie2 was the imaging parameters K^{trans} and IAUC, as shown in Figure 3.

Reviewer comment: thanks

There are a number of areas where the data needs to be presented more clearly:

- lack of N's of data points / patients actually contributing to the analysis at each point
The requested patient numbers have been added in sFigure 1.

Reviewer comment: thanks

- The authors need to explain the rationale for any pre-specification of any of their groupings of focus (e.g. why did analysis focus on pTie2, out of how many choices [line 521] etc)

We focused on plasma Tie2 because (1) the correlation network analysis demonstrated that Tie2 was the only protein that increased its correlation with other biomarkers, including the imaging parameter, K^{trans} , following bevacizumab; (2) our previous publication in ovarian cancer highlighted pTie2 as a predictive and a response biomarker. We have addressed this in detail in the revised manuscript (page 5, last paragraph).

Reviewer comment: thanks

- The pTie2 association with progression was published previously for Ovarian Cancer in Zhou et al 2016 BJC, how does the data compare?

Here, in colorectal cancer, we have confirmed that pTie2 is a response biomarker for bevacizumab. With data from MRI imaging we have now demonstrated that the pTie2 signal is derived from the tumour vasculature. In both studies we have shown that changes in pTie2 concentration are only induced by bevacizumab (rather than cytotoxic chemotherapy). In both studies, the definition of vascular response is a greater than 25% reduction in plasma concentration of pTie2. However, the definitions of vascular progression in ovarian and colorectal cancer were a 50 and 30% increase above the nadir in pTie2, respectively. We can expand further on this issue if needed in the discussion.

Reviewer comment: Expanding on this would be useful as it help orientate readers as to how this new analysis fits with other work.

We have split the last sub-section of the original manuscript results section into two parts to address this point. The first is the cohort level validation and the second, which addresses the reviewer's question, concerns precision medicine and how to use the biomarkers when treating a patient. We have also expanded the discussion quite significantly to discuss the utility of the biomarker in the clinic, its validation and potential weaknesses of the studies so far.

- The wording used to describe potential associations should be softened.
Revised (pages 10 and 11).

Reviewer comment: thanks

- Visual of actual data to demonstrate "inflection point" idea would be helpful. Fig S2 (all patients, not selected to progressors), CK18 has inflection but Tie2 looks more like VEGFa?
In the results section entitled "*The pTie2 signature is derived from tumor vasculature*", we show that pTie2 trajectories are heterogeneous but can be clustered into two distinct patterns in which the inflection point occurs in the patients' time course (Figure 3). The U shaped inflection curves disappear when visualising patients as a whole (the reviewer's point), as shown in sFigure 1, because the heterogeneous trajectories cancel each other out. CK18 trajectories are more homogeneous whether they are clustered (Figure 3) or not (sFigure 1).

Reviewer comment: is it possible to provide a couple of examples as to how the population analysis translate to a an individual patient, which is where the application of the work really is.

This was a very helpful suggestion and examples have been included in figure 6. The legend explains how to interpret the graphs for each patient.

- Fig 2 while there is logic behind the %PFS axis but how does that relate to the visit windows where data are collected, are the points actually at 20,40,60,80%, are data moved to the nearest one, etc.
Data points were moved to the nearest relevant time window when calculating %PFS. Multiple data points belonging to the same window were averaged. These have been described in detail in the Methods section.

Reviewer comment: thanks

- Fig 2, what are the cut-offs selected, how many pts in each?

In this study we avoided comparing patients using pre-specified or data-driven cut-offs so that data were not over-fitted. Specifically, the unsupervised clustering analysis was applied to cluster pTie2 longitudinal profiles and no cut-off was pre-specified.

Reviewer comment: thanks

Is done on pre-treatment value which is important.

Uni- and multi-variate analyses of all clinical and biomarker parameters were performed before treatment and this identified the K^{trans} : VEGFR2 ratio as having independent prognostic significance in addition to conventional prognostic factors for colorectal cancer. We hypothesised that the hyper-acute reduction of biomarker levels represented the treatment effect of bevacizumab.

Reviewer comment: Ok clear

Clear drop in all patients in first period, so if this is a real finding, it is only valid assuming the patient is PFS-free for xx days?

The hyper-acute reduction in pTie2 was due to the effects of single agent bevacizumab. In keeping with standard international clinical practice, the first potential assessment of efficacy is after 6 weeks of treatment and the first date of progression was day 46, in keeping with this concept. However, the progression free intervals were much longer than the hyperacute reduction in pTie2, as detailed in the manuscript.

Reviewer comment: this is clear

- Fig 3a choice of variable and cut-off, as in many others, presumably selected from a larger list of choices and so not surprising to find one. Could give the medians, looks like they would not be so different. The flat section pre-200 days is odd, scans were @12w and the steps restart thereafter? How did the early progressor fall into these timepoints.

Over-fitting and multiple comparisons are typical statistical challenges in biomarker studies. In this study the medians were used instead of data-driven ones to avoid the pitfall of over-fitting data. We have also applied an internal validation method named bootstrap, which allows 95% CI of p-values to be derived and the impact from outlying data points to be eliminated. These approaches are consistent with the REMARK guidelines for biomarker discovery. The data show that there was no progression between 100 and 189 days after treatment. Accordingly a flat area in the survival curves is observed. All progression events were illustrated in the Kaplan-Meier curves.

Reviewer comment: could the bootstrap methodology be described in the methods?

The bootstrap methods are outlined in the legend to table 3, described on pages 5 and 35 and a reference has been included (ref 47).

- Fig 4 While successful prediction uses a 30%/50% increase from nadir it is not clear what value this would relate to.

Use of the nadir value for subsequent clinical comparisons, as proposed here, is directly analogous with the way that we use Ca-125 to guide therapy according to the international guidelines in gynaecological oncology. The nadir is patient-specific and oncologists are used to this concept.

Reviewer comment: it would be useful to elaborate a little more as to how this actually deploys for plasmaTie2 with specific concentrations and examples.

Again, this was a very helpful comment, which resulted in individual patient data being shown in figure 6. The legend explains how to interpret the individual patient biomarker data.

- Fig 5 is a hypothetical visual, not based on data, hugely suggestive without evidence any observed changes are caused by bev, the hypothetical nature of this should be clearly highlighted.

We agree with the reviewer that this is hypothetical. However, the data in the paper are complicated and we felt that a diagrammatic summary would be helpful for the reader. Please note that we feel the reviewer's statement that the "visual [is] hugely suggestive without evidence any observed changes are caused by bev" is not correct. Much of the manuscript demonstrates the impact of single agent bevacizumab on tumour biomarkers and growth.

If there were a specific point that the reviewers/editors would like us to change that would be completely acceptable. Indeed, the figure could be deleted. It was simply incorporated to act as a summary of the whole manuscript.

Reviewer comment: reviewer or editor would need ot comment. The diagram is not that helpful, but a schematic would help.

This has now become figure 6 as a result of other requests by reviewers. Hopefully the schematic

does help now as it is followed by real patient data, showing similar concentration-time data.

- Fig S1 Not clear what “Slightly lower overall survival statistics” are, relative to what?

This phrase refers to the range of reported overall survival statistics for patients with metastatic colorectal cancer, which have been described in the literature. We wanted to demonstrate that the overall survival statistics for our patients were within the range of published international survival statistics but that our survival statistics were towards the lower end of that range because we had to recruit patients who had at least one lesion that was amenable to advanced MRI and therefore had large tumours of between 3-10 cm diameter. We can add more references to this phrase if required by the journal but it would add to the number of references, if permitted.

Reviewer comment: addition of a few words to the text would be sufficient to resolve the point.

We have added comments along these lines into the legend of figure 1.

- Need to make it clear how many patients data points feed into each analysis and how the cut offs for each group were determined

As mentioned in response to previous questions we have added the number of patients' data points to sFigure 1. We did not use cut-offs for grouping patients.

Reviewer comment: clear

- What defines progression?

Progression was defined according to RECIST 1.0, as described in the methods section. Reviewer comment: clear

Minor point

pTie2 implies phospho-Tie2 this could be confusing?

We understand this point and can change to an alternative format if needed. However, this was the briefest format and was therefore most appealing.

Reviewer comment: Still confusing and alternate phrase even plasma-Tie2 would help

We have changed pTie2 to just Tie2 throughout the manuscript.

How do the authors feel this work could be implemented?

This very important issue has been addressed in the re-written manuscript in which we demonstrate that bevacizumab is associated with a 75% vascular response rate, as defined by a 25% reduction in pTie2. Vascular progression is defined as a 30% increase in pTie2 over nadir. From a pragmatic point of view in the future we propose that a 50% increase in pTie2 over the nadir could be broadly implemented as a definition of vascular progression.

Reviewer comment: The paper would benefit from representing proper values in addition to the percentages

In response to the question in black text we have made extensive changes to address this issue in that we have introduced a results section on precision medicine; we have added individual patient data graphs and we discuss implementation in the discussion.

In response to the question in red, the third column of table 2 provides the concentration or value of all biomarkers +/- the standard deviation allowing readers to work out absolute values. We will also publish the entire data set through the University of Manchester, if the manuscript is accepted for publication.

Referee #3 (Remarks to the Author): General comments:

This manuscript describes an interesting study examining the potential utility of plasma Tie2 as a biomarker of tumor vascular response to bevacizumab anti-VEGF therapy in patients with metastatic colorectal cancer. It is a natural extension of previous work that this same group published in May 2016 in the British Journal of Cancer (ref #6), which identified plasma Tie2 as a biomarker of vascular progression in patients with ovarian cancer receiving bevacizumab. [Note: I cannot provide a technical assessment of the complex mathematical modelling presented, but the 2016 BJC study appears to have used very similar methodology.] Unfortunately, one of the authors' main claims – that pTie2 correlated with PFS and tumor response – was not strongly supported. See point#4 below for suggestions of additional analyses that might directly test these correlations and potentially strengthen this claim.

We will address that point below.

Major issues:

1. Lines 499-502 noted that the analyses of MRI data, to generate parameters such as K_{trans} , were confined to each patient's largest lesion. Lines 386-387 imply that the largest lesion happened to be a liver metastasis (3-10cm) for most patients. Table S1 currently summarizes the % of patients with metastatic disease sites in bowel (83%), liver (83%), lung (36%), LNs (34%),

etc. and primary tumor in situ (31%). Can authors also add to this table the distribution of what the “largest lesion” meant for the 70 patients?

In brief, we take a pragmatic approach to tumour selection for imaging, which is summarized as follows. Through multiple publications and establishment of Standard Operating Procedures we have established imaging protocols for patients whose tumours measure at least 3 cm so that should the tumour respond to treatment, sufficient voxels remain available for quantitative analysis; on the other hand, tumours should be less than 10cm in cranio-caudal dimension so that whatever size-related changes occur, the entire tumour can be quantitatively imaged within one MRI frame. Liver has become a preferred site of imaging in sequential studies like this as the imaging protocols are reliable and reproducible with good tissue contrast. We have established breath hold protocols to ensure repeated imaging is feasible, reproducible while minimizing movement.

Some patients had several tumours and we have written a further technical manuscript on the comparison between tumours in an individual vs between patients, which demonstrate the similar behaviour of tumours within patients. This manuscript will be attached to the cover letter to the editor.

Reviewer comment: The manuscript would still benefit from adding in the requested information, unless the paper mentioned is in press then it can be referenced?

The revised manuscript now contains data that should address the reviewer’s request. Table 2 column 3 on page 18 shows the median volumes of patients’ tumours +/- SD. Figure S3 (page 56) shows the pre-treatment variance in all biomarkers including whole tumour volume.

The host vasculatures in these different organ sites can obviously differ in many ways and in turn affect the relative role of angiogenesis vs. vessel co-option in supporting metastatic tumor growth. Do the 2 distinct patterns of pTie2 trajectories (Fig. 2a) reflect considerably different distributions of organ sites?

Unfortunately, we cannot address the relative contribution of angiogenesis and vessel co-option as imaging protocols currently do not distinguish the two mechanisms.

A chi-square test of independency has shown that there is no significant association between tissue origin of lesion and pTie2 trajectories. Similarly, no association was found between PS, sex and pTie2 indicating that patterns in pTie2 are more likely to relate to treatment response of individual. The manuscript has been revised accordingly.

Reviewer comment: adding in a comment to the discussion on the contribution of multiple lesions to the signal would be valuable and reflecting the potential for different vascular beds to respond differently.

We have added a paragraph on page 15 that describes some caveats for the study. In that

paragraph we talk about the potential confounding effects of simultaneous evaluation of different biomarkers. We feel that this is probably an appropriate balance of commentary since all imaging studies of effective VEGFis have shown imaging effects on K^{trans} or $IAUC_{60}$ no matter which tumour site was imaged. Further, we have shown a strong correlation between WTV (single tumour behaviour) and CK18, a marker of total tumour burden.

2. Fig. S3: The legend says that the circulating biomarkers on the left and imaging biomarkers on the right were measured at different timepoints – hence the dotted line separating the left vs. right sides and the inability to assess any correlations between the 2 sides. Instead of labeling the whole figure as “<3 month”, can authors state exactly what was the timepoint used for each side? Why couldn’t this analysis be done at the day-22, 6-week, 6-month mark, when both circulating biomarkers and imaging biomarkers were measured, according to Table S5?

In sFigure 3 (sFigure 2 in the revised manuscript) the network for circulating biomarkers was created using data collected following week 2 and extending to week 12. These data were then compared with imaging data from week 12. While we could limit the data to week 12 only, this would remove preceding circulating biomarker data and in any case no significant differences in the networks were observed after week 2.

Reviewer comment: I am not sure that the reply really answers what they reviewer was looking for – I read the question as asking for the continuous parallel for the biomarkers at the same time points over the course of the study.

In accordance with the reviewer’s request we have limited the data presentation to the same time period for both circulating and imaging biomarkers. We have therefore removed the dotted line separating the two biomarker data sets allowing comparison between the circulating and imaging biomarkers. Please note that this has become Figure S4.

This is important because on lines 116-117, authors claim that the addition of cytotoxic chemotherapy to bevacizumab (beginning on day 15) did not significantly impact the correlations between circulating and imaging biomarkers that emerged after bevacizumab monotherapy as assessed on day 14 (i.e., Tie2 to K^{trans} & Tie2 to $IAUC$ in Figure 1b). How can this claim be supported when no correlations are shown between the twosides in Fig. S3?

Correlations within the individual circulating and imaging networks demonstrated few changes following treatment with cytotoxic chemotherapy and bevacizumab, whereas the

administration of single agent bevacizumab had a profound impact on the correlations.

Reviewer comment: I assume the authors mean that the correlations of these biomarkers don't change further on the addition of chemotherapy. The disconnect in Tie-2 change on bev alone and the CK18 change on addition of chemotherapy is nice to see. May be worth pulling out some of the biomarker changes in the discussion. Which markers tell you about the vasculature and blood flow, and which about impact on the tumour cell compartment would be valuable.

The point made by the reviewer in black text has now been addressed through a single time point analysis, which is reported in Figure S4, confirming that cytotoxic chemotherapy does not impact further on the biomarker networks.

With reference to the points made in red, we have published many papers on imaging the vasculature but feel that a detailed discussion of each of the imaging parameters (i.e. discussing flow vs perfusion vs permeability) would obscure the message here. However, we agree with the reviewer about understanding the epithelial vs vascular compartment message and have illustrated this in a schematic in Figure 6a and in individual patient data in Figures 6b-e. The results section on precision medicine describes how we would suggest the biomarkers are used in the clinic.

3. Fig 1b and lines 127-128: how did the authors choose to focus on K^{trans} rather than IAUC once both were identified as emergently correlated with Tie2 after bevacizumab?

K^{trans} and IAUC are very similar in concept and mathematically. Compared with IAUC, K^{trans} demonstrated a slightly larger correlation (0.47 vs. 0.45) but either of them could be used as a biomarker.

Reviewer comment: I think the reviewer may have been implying that both should be shown

?

Please note that both K^{trans} and IAUC have been studied as pharmacodynamic endpoints for VEGFi. Both have shown utility and here K^{trans} had a slightly stronger correlation as a result of bevacizumab treatment. Incorporation of IAUC instead or as well would not change the message here. In addition, both pertain to the vasculature and the point of their incorporation was to relate circulating biomarker data to tumour vascular imaging data. Thus, the conclusions would be the same whichever of these imaging biomarkers were included.

4. The authors' key claim in this paper is essentially summarized in lines 155-161: "Taking these data together, we have identified a group of patients with significantly worse PFS, whose tumours are characterised by high VEGF-R2:Ktrans ratio before treatment, by a lack of tumour size reduction after two weeks' single dose of bevacizumab and by transient bevacizumab-induced vascular control that was reflected by less profound and less durable reductions in pTie2. Thus the trajectory and duration of reductions in Tie2 is significantly associated with response (volumetric change) and PFS." However, the logic behind this claim is shaky. This seems to be a case of showing $A \rightarrow B$ and $B \rightarrow C$, then concluding $C \rightarrow A$, except $B \rightarrow C$ is weak. First, higher pre-Bev "VEGFR2:Ktrans" is shown to correlate with reduced PFS (Fig 3a) and reduced tumor volumetric response (Fig. 3b-c). Next, authors showed that patients with high "VEGFR2:Ktrans" were more likely to have transient pTie2 reductions than patients with low "VEGFR2:Ktrans" (Table S4) – which is technically true (47.5% vs. 23%), but this still means that 52.5% of patients with high "VEGFR2:Ktrans" had sustained pTie2 reductions! In other words, the subgroup of patients with higher "VEGFR2:Ktrans" (and reduced PFS) is almost equally likely to have transient pTie2 reductions vs. durable pTie2 reductions.

The reviewer is correct that subgroups with high VEGFR2:K^{trans} are equally likely to have transient and sustained pTie2 reductions. However the analysis shows that patients with high VEGFR2:K^{trans} are significantly more likely to have transient pTie2 reductions than patients with low VEGFR2:K^{trans}. In other words we are comparing two subgroups of patients, rather than different types of pTie2 reduction.

Reviewer comment: fine

So, instead of going about this indirectly, why don't the authors directly analyse the link between pTie2 and WTV (i.e., plotting blue "transient pTie2" vs. black "durable pTie2" curves for x =whole tumor volume and y =%FPS, as an extra panel D in Fig.2) and the link between pTie2 and PFS (i.e., plotting "transient pTie2" vs. "durable pTie2" Kaplan-Meier curves, as an extra panel D in Fig.3)?

There are several reasons why this is not the best way to address this question:

- 1) We have tried to avoid arbitrary cut-offs that would incur the error of over-fitting of data. Our approach has been to use unsupervised analysis to avoid these issues.
- 2) In an analysis of PFS, there are several clinical and biomarker factors, which we identify at the start of the manuscript that are associated with outcome. Indeed one of the major outputs of the paper is that we can better model progressive disease when we bring epithelial and vascular biomarker data together. Thus the proposed analysis would ignore all of these other important factors.
- 3) Finally, WTV refers to the specific lesion that we were studying by advanced MRI rather than total tumour burden.

Reviewer comment: again, not sure this answer what the reviewer was actually asking, but the answer raises the question about whether progression in general was determined by a change in the target lesion or another lesion. If another lesions progress then does it change

the conclusions driven from the later imaging data?

There are two questions here: The reviewer who used black type asks about comparing WTV and pTie2. However, that question misses the point of the paper, which is to argue that we see a much stronger relationship between biomarkers and progressive disease when we model epithelial (CK18) and vascular (Tie2) biomarkers together. Using Tie2 alone results in inferior modelling.

The question in red text concerns the definition of progressive disease. In accordance with standard clinical practice and trial regulations progressive disease was defined according to RECIST 1.0 (page 28). It would have been unethical to stop treatment on the basis of changes in a single lesion. That said, the marker of total tumour burden, CK18, correlated strongly with single tumour measurements (WTV; page 5).

5. From Figure 4, the conclusion was made that the two biomarkers, CK18 and Tie2, combined makes a better predictor of progressive disease than either alone. To illustrate pTie2's greater superiority over CK18, authors quoted an average lead-time of 50 days in pTie2's ability to predicting PD. Can authors also state CK18+pTie2's lead time over pTie2 alone? Because in Fig. 4a/b, CK18+pTie2's accuracy often doesn't appear to be that much higher than that of pTie2 alone, which compromises the argument for the added utility of multi-compartment modeling.

The inference that lead-time will be improved by adding CK18 to pTie2 together is not correct. We have shown that vascular progression (pTie2) usually occurs before epithelial progression (CK18), presumably because vascular progression, i.e. improved blood and nutrition, supports subsequent epithelial progression. This does not imply that pTie2 is more useful than CK18. The advantage of modelling pTie2 with CK18 is that it improves prediction accuracy, reflecting the need to consider tumour vascular and epithelial progression together.

Reviewer comment: agree with the rationale, however the answer raises the question as to whether addition markers would improve the strength of the prediction.

We have examined the issue of composite biomarkers, as suggested by the reviewer (red text). This point was also made by another reviewer. We find that Ang2*Tie2 is very slightly superior to Tie2 alone. However, Ang2 is much less stable as a biomarker and the increment in performance is so small that it does not make a significant difference and we therefore recommend focusing on Tie2 alone.

6. The final paragraph (lines 194-203) is quite confusing. How exactly are authors proposing

that the combined CK18+pTie2 biomarker be used clinically to guide the use of anti-angiogenic therapies? Are authors suggesting that once patients reach “CK18+pTie2”-predicted vascular progression, it might be worth switching them to alternate anti-angiogenic therapies, e.g., from an anti-VEGF agent to Ang2/Tie2-targeted agent? Are authors suggesting that patients should be maintained on their existing anti-angiogenic therapy when they reach RECIST-based PD if they have not yet reached “CK18+pTie2”- predicted vascular progression? Figure 5 is also confusing – what do the illustrative U- shaped curves mean?

We agree with the reviewer and in our completely rewritten manuscript we have addressed this issue. There are two critical outputs:

- 1) We have shown that bevacizumab is associated with a 75% vascular response rate in colorectal cancer, where a response is defined as a greater than 25% reduction in pTie2 and progression is defined as a greater than 30% increase in pTie2. We are confident that these findings are genuine and ready for clinical use because of our previous ovarian cancer study and because of other published pharmacodynamic data with pTie2 observed in patients with colorectal and gallbladder cancer and glioma who were treated with the VEGF RTKi, cediranib. These definitions of vascular response and progression now allow clinicians to use VEGF inhibitors optimally for the first time. We can tell our patients when the drugs are working and when they stop working. We also know from our ovarian trial and from additional work in colorectal cancer that cytotoxic chemotherapy alone (manuscript available) does not impact on pTie2 thus our vascular response/progression definitions apply whether VEGFi are given alone or in combination. Anatomical CT scanning is not irrelevant. However, when deciding whether to stop a course of VEGFi, we can then merge, in the clinical sense rather than mathematically, radiological information together with pTie2 data, as we do with conventional tumour markers, to decide how best to manage a patient with indolent or small volume progressive disease.
- 2) The other critical output is that by demonstrating the additive clinical value of modelling epithelial and vascular information together and by taking other 1st-line, 2nd line and 3rd-line/chemo-resistant VEGFi trial data together in ovarian and colorectal cancer, our results imply that we should treat the vascular compartment at each recurrence using pTie2 as the guide for oncologists.

Reviewer comment: I am not sure that the point raised by the reviewer is fully dealt with, however despite this these points should be added to the discussion?

We hope that these issues have now been addressed through the inclusion in the results section of a subsection entitled “Precision medicine...”. This subsection, together with the individual patient data shown in Figure 6 should explain how the biomarkers should be used in practice.

Figure 5 is a diagram that we hoped would summarise the message from the paper in a single figure. It is not a data diagram and therefore could be removed. However, as the paper is complex, we wanted to summarise the manuscript for readers in one figure.

Reviewer comment: the diagram is not that helpful is not that useful but some schematic is required.

This is now figure 6a. I hope that the diagram has become more useful as it depicts in a more straightforward manner what we are seeing in the individual patient data in Figures 6b-e.

Minor issues:

- Fig. S1 legend: What do authors mean by “The slightly lower overall survival statistics, while not statistically worse...”? Lower than what? Unclear what the comparison is against when panel b is only showing one OS curve.

This has been explained above in response to the other reviewer’s question.

- Line 149: Wrong statistics quoted here for Fig. 3c. “(median PFS 248 vs. 348 days, $p=0.0008$)” refers to Fig. 3a several lines above.

Revised

- Line 154: Might be clearer if “...transient reductions in circulating Tie2 (Fig 2, Fisher’s exact test $p=0.014$, Table S4)” is changed to “...transient reductions in circulating Tie2 as defined in Fig 2 (Fisher’s exact test $p=0.014$, Table S4)”.

Revised

- Line 423: “(Supplemental Table 1)” doesn’t seem to be the correct reference.

Checked and revised

- Line 426: Aren’t circulating biomarkers also measured every 6 weeks until PD, according to Table S5 footnote #3?

Yes the circulating biomarkers are measured every 6 weeks.

- Line 555: please briefly summarize explain “censored events” (e.g., lost to followup?) that were lumped as “progression”

Only 2 patients fell into this category and they were both long progressors (PFS>3 years).

They were censored due to trial closure and so will not cause bias.

- Reference style is very inconsistent. Some journals are abbreviated and some not. Some DOI’s are underlined and some not. Multiple references included “[journal name]: official journal of [name of Society/Organization]” – journal name is sufficient. Ref#4 has an extra “.“ between Lancet Oncology. Ref #17 is missing journal name. Reference 6 and 7 are duplicates.

Revised.

- Line 615: MCMC was never defined in the manuscript (Markov Chain Monte Carlo?)
This has been properly defined in the revised manuscript.

Reviewer comment: All fine

REVIEWERS' COMMENTS:

Reviewer #2 (Remarks to the Author): The revised manuscript from Jayson et al is much improved. The new representations of the data and the added detail (e.g. examples of individual patients) make the work much more accessible. The added description and enhanced discussion are also very usefu